# GRQA: Global River Water Quality Archive

Holger Virro[1], Giuseppe Amatulli[2,3], Alexander Kmoch[1], Longzhu Shen[4,5], and Evelyn Uuemaa[1]

[1]Department of Geography, Institute of Ecology and Earth Sciences, University of Tartu, Vanemuise 46, Tartu, 51003, Estonia
[2]Yale University, School of the Environment, New Haven, CT, 06511, USA
[3]Yale University, Center for Research Computing, New Haven, CT, 06511, USA
[4]HyperAmp, Barnwell Road, Cambridge CB5 8RQ, UK
[5]Spatial-Ecology, Meaderville House, Wheal Buller, Redruth, TR16 6ST, UK

**Correspondence:** Holger Virro (holger.virro@ut.ee)

**Abstract.** Large-scale hydrological studies are often limited by the lack of available observation data with a good spatiotemporal coverage. This has affected the reproducibility of previous studies and the potential improvement of existing hydrological models. In addition to the observation data itself, insufficient or poor quality metadata has also discouraged researchers to integrate the already available datasets. Therefore, improving both, the availability, and quality of open water quality data would
increase the potential to implement predictive modeling on a global scale.

The Global River Water Quality Archive (GRQA) aims to contribute into improving water quality data coverage by aggregating and harmonizing five national, continental and global datasets: CESI, GEMSTAT, GLORICH, WATERBASE and WQP. The GRQA compilation involved converting observation data from the five sources into a common format and harmonizing the corresponding metadata, flagging outliers, calculating time series characteristics and detecting duplicate observations from sources with a spatial overlap. The final dataset extends the spatial and temporal coverage of previously available water quality
data and contains 42 parameters and over 17 million measurements around the globe covering the 1898–2020 time period. Metadata in the form of statistical tables, maps and figures are provided along with observation time series.

The GRQA dataset, supplementary metadata and figures are available for download on the DataCite and OpenAire enabled Zenodo repository https://doi.org/10.5281/zenodo.5097436 (Virro et al., 2021).

## 1  Introduction

Human-driven loads of nutrients to aquatic ecosystems have become the main driver of eutrophication in waterways and coastal zones (Desmit et al., 2018; Sinha et al., 2019). Agricultural production is already one of the major forces behind environmental degradation (Foley et al., 2011), and population growth is increasing that pressure (Mueller et al., 2012). The use of nitrogen (N) and phosphorus (P) fertilizers to increase agricultural productivity is predicted to increase threefold by 2050 unless more
efficient fertilizer use can be implemented (Tilman et al., 2001). At the same time, it has been estimated that "globally, over 3 billion people are at risk of disease because the water quality of their water source is unknown, due to a lack of data" (UN-Water, 2021). In order to achieve the UN SDG 6, we need better understanding of our water resources and water quality. Monitoring and modeling the hydrochemical properties of rivers is essential for understanding and mitigating water quality

deterioration due to agricultural and industrial non-point source pollution (Krysanova et al., 1998; Leon et al., 2001; Wu and
Chen, 2013). Modeling of different water quality indicators such as nutrients (Caraco and Cole, 1999; He et al., 2011), carbon
compounds (Evans et al., 2005; Hope et al., 1994), sediments (Choubin et al., 2018; Ouyang et al., 2018) and oxygen (Radwan
et al., 2003; Singh et al., 2009) gives valuable understanding of hydrochemical cycles and enables to estimate the effect of
human influence on them.

Traditional approaches to water quality modeling consist of applying bottom-up, physically based models on the catchment
level (Wellen et al., 2015). Calibration and validation data in the form of water quality observations used when developing the
model and verifying its performance is usually gathered through *in situ* observations and, more recently, automated sensor net-
works. Although airborne remote sensing based data acquisition methods have been successfully used to supplement field data
for lakes (Chen and Quan, 2011; Toming et al., 2016), applying those methods is only viable in the case of rivers with a large
enough surface area (Olmanson et al., 2013). Therefore, improving the river water quality data spatial and temporal coverage
with remote sensing is limited. Significant progress has been made in improving the technical capabilities and lowering the
installation and maintenance costs of the field sensors, but the spatial and temporal coverage of observation sites remains to be
an issue (Pellerin et al., 2016).

In order to improve the spatial coverage of water quality and hydrological data, different solutions have been used in pre-
dictive hydrological mapping. Until recently, a common approach for predicting water quality and hydrological phenomena in
ungauged catchments has been the application of already existing process-based models to catchments with similar character-
istics (Hrachowitz et al., 2013; Strömqvist et al., 2012; Wood et al., 2011). These physical models usually require extensive
calibration along with location-specific knowledge, which limits the wider applicability and spatial upscaling that can be done
(Abbaspour et al., 2015; McMillan et al., 2012).

Recently, advances in implementing machine learning (ML) methods in hydrology have given rise to a new, data-driven
approach to hydrological modeling (Mount et al., 2016). Comparison of physically based and ML approaches has shown that
ML methods can achieve a similar accuracy to the physically based ones and outperform them when describing nonlinear
relationships (Chau, 2006; Ouali et al., 2017; Papacharalampous et al., 2019). The recent advent of so-called physics-guided
ML, which entails combining process-based models with ML methods is likely to become more applicable in the near future
as well (Kratzert et al., 2019; Shen et al., 2018; Marzadri et al., 2021).
Nevertheless, a major problem related to large-scale predictive hydrological modeling has been the lack of available obser-
vation data with a good spatiotemporal coverage (Bierkens, 2015). This has affected the reproducibility of previous studies
and the potential improvement of existing models (Blöschl et al., 2019; Meals et al., 2010; Stagge et al., 2019). In addition
to the observation data itself, insufficient or poor quality metadata has also discouraged researchers to integrate the already
available datasets. Here, ambiguities in supplementary metadata such as parameter names, units and methods of measurement
has limited the use of open data for large-scale water quality modeling purposes (Archfield et al., 2015; Hutton et al., 2016;
Sprague et al., 2017). Therefore, improving both the availability and quality of open water quality data would increase the
potential to implement predictive modeling on a global scale. Global ML models have been already successfully used for
discharge modeling (Beck et al., 2015; Gudmundsson and Seneviratne, 2015) and recent years have seen the publication of

global discharge datasets (Do et al., 2018; Harrigan et al., 2020). The publication of global and continental datasets (Hartmann et al., 2014; Read et al., 2017) could make ML methods applicable for large-scale water quality modeling as well (Shen et al., 2020). However, issues related to a lack of training and validation data due to general data scarcity affects model accuracy and, therefore, limits the further adoption of ML for global water quality predictions (Chen et al., 2020).

We aim to address the aforementioned issues by presenting the novel Global River Water Quality Archive (GRQA) by integrating and harmonizing five different global and regional datasets. The resulting dataset has combined observation data for 42 different forms of some of the most important water quality parameters relevant for nutrients (e.g. water temperature, oxygen, phosphorus, nitrogen and carbon compounds). Supplementary metadata and statistics are provided with the observation time series to improve the usability of the dataset. An extensive data catalogue with maps showing the spatiotemporal coverage and graphs describing the distribution of all 42 parameters as supplementary material of the study (see Supplement). We report on developing a harmonized schema and reproducible workflow that can be adapted to integrate and harmonize further data sources. In addition, we provide recommendations for improving multi-source water quality data compilation, especially focusing on the metadata quality and adhering to the FAIR Data Principles (Wilkinson et al., 2016). We conclude our study with a call for action to extend this dataset and hope that the provided reproducible method of data integration and metadata provenance shall lead as an example.

## 2 Data

A total of five data sources were used to compile the GRQA with two being global, one regional, and two national level (Table 1). All datasets with the exception of GEMSTAT are publicly available to download online as CSV or Excel file packages. GEMSTAT data can be requested via email. The number of available observation sites was highly dependent on the source with the Water Quality Portal (WQP) maintained by the United States Geological Survey (USGS) having the most sites. Files used during the creation of GRQA are listed in Table 2.

### 2.1 CESI

The first dataset included in GRQA originated from the Canadian Environmental Sustainability Indicators program (CESI) operated by Environment and Climate Change Canada (ECCC), which is a Canadian governmental department responsible for coordinating environmental policies and programs. CESI consists of water quality measurements collected by federal, provincial and territorial monitoring programs from Canadian rivers from the 2002–2018 time period (Environment and Climate Change Canada, 2020). CESI data is mainly focused on heavy metals, so out of the 42 of parameters included in GRQA only eight were available in CESI (Table 1). It is the smallest of the five source datasets with site count ranging from two to 77 per parameter. Mean time series length per site is approximately 13 years and the average number of observations per site is 145.

**Table 1.** Source datasets used for compiling GRQA with their total number of observations, parameters and timeframe length in GRQA. All datasets were retrieved on November 16, 2020.

| Dataset | Name | Data provider | Observations | Timeframe | Parameters (source/ **GRQA**) | Site count range | Mean time series length per site | Mean observation count per site |
|---------|------|---------------|--------------|-----------|-------------------------------|------------------|----------------------------------|----------------------------------|
| | | | *n* | | *n*/***n*** | *n* | *years* | *n* |
| CESI | Water quality in Canadian rivers | Environment Canada | 30,457 | 2002–2018 | 8/**42** | 2–77 | 12.9 | 145 |
| GEMSTAT | Global Freshwater Quality Database | International Centre for Water Resources and Global Change | 2,094,598 | 1950–2020 | 32/**42** | 7–4,274 | 9.2 | 77 |
| GLORICH | GLObal RIver Chemistry database | Institute of Geology of the University of Hamburg | 3,231,797 | 1942–2011 | 26/**42** | 4–9,728 | 4.1 | 41 |
| WATERBASE | Waterbase - Water Quality | European Environment Agency | 306,332 | 2008–2018 | 15/**42** | 4–1,976 | 1.4 | 19 |
| WQP | USGS Water Quality Portal | Environmental Protection Agency | 8,689,335 | 1898–2020 | 37/**42** | 1–59,000 | 3.4 | 25 |

## 2.2 GEMSTAT

The Global Freshwater Quality Database GEMStat (Färber et al., 2018) is hosted by the International Centre for Water Resources and Global Change (ICWRGC) and provides inland water quality data within the framework of the GEMS/Water Programme of the United Nations Environment Programme (UNEP). GEMStat contains over 7 million samples from approximately 5,700 sites in 75 countries. The data was obtained through a custom request to their data portal (International Centre for Water Resources and Global Change, 2020).

Approximately 500 water quality parameters were available in the GEMSTAT database, out of which 32 were used when compiling GRQA (Table 1). Observations cover the period 1950–2020 and mean observation count per parameter is approximately 41. Mean time series length per site is nine years. Site count per parameter ranges from less than ten (dissolved and total carbon) to 4,274 (total phosphorus).

## 2.3 GLORICH

The GLObal RIver CHemistry (GLORICH) database (Hartmann et al., 2014) is a collection of hydrochemical data from more than 1.27 million observations and more than 18,000 sampling locations across the globe. The samples originate from various environmental monitoring programs and scientific literature.

**Table 2.** Source dataset files used for compiling GRQA. WQP sites and observations were downloaded separately for each parameter and file names were assigned during the process.

| File name | Size (MB) | Rows | Description | Sheet name | Source |
|---|---|---|---|---|---|
| wqi-federal-raw-data-2020-iqe-donnees-brutes-fed.csv | 171.5 | 314,867 | Observation data | | CESI |
| data_request.xls | 2.4 | 5,419 | Site data | Station_Metadata | GEMSTAT |
| data_request.xls | 2.4 | 30 | Parameter data | Parameter_Metadata | GEMSTAT |
| data_request.xls | 2.4 | 311 | Method data | Methods_Metadata | GEMSTAT |
| pH.csv | 21.9 | 372,211 | Observation data | | GEMSTAT |
| Carbon.csv | 19.2 | 337,928 | Observation data | | GEMSTAT |
| Nitrogen.csv | 65.1 | 1,052,823 | Observation data | | GEMSTAT |
| Phosphorus.csv | 24.3 | 386,113 | Observation data | | GEMSTAT |
| Oxygen_Demand.csv | 20.1 | 331,617 | Observation data | | GEMSTAT |
| Solids.csv | 11.8 | 201,628 | Observation data | | GEMSTAT |
| Water_Temperature.csv | 23.9 | 370,335 | Observation data | | GEMSTAT |
| Oxygen.csv | 30.6 | 488,749 | Observation data | | GEMSTAT |
| Sampling_Locations_v1.shp | 0.4 | 15,553 | Site point data | | GLORICH |
| sampling_locations.csv | 1.6 | 18,897 | Site name data | | GLORICH |
| catchment_properties.csv | 10.2 | 15,514 | Catchment data | | GLORICH |
| hydrochemistry.csv | 273.3 | 1,274,102 | Observation data | | GLORICH |
| Waterbase_v2019_1_S_WISE6_SpatialObject_DerivedData.csv | 15.1 | 62,288 | Site data | | WATERBASE |
| ObservedProperty.csv | 0.2 | 888 | Observation data | | WATERBASE |
| Waterbase_v2019_1_T_WISE6_DisaggregatedData.csv | 10019.2 | 39,121,790 | Observation data | | WATERBASE |
| WQP_*_sites.csv | 2543 | 9,467,369 | Site data | | WQP |
| WQP_*_obs.csv | 2749.8 | 10,088,212 | Observation data | | WQP |

Out of 47 water quality parameters available in the raw data, 26 were chosen to be included in the GRQA (Table 1). The samples cover the time period of 1942–2011, but the length of the time series is dependent on the parameter. Mean time series length per site is less than a decade for all parameters. The number of available sites per parameter ranges from just four (particulate organic nitrogen) to 9,728 (dissolved inorganic phosphorous). The dataset can be downloaded at Pangaea (Hartmann et al., 2019).

## 2.4 WATERBASE

Waterbase is the generic name given to the European Environment Agency's (EEA) databases on the status and quality of Europe's rivers, lakes, groundwater bodies and transitional, coastal and marine waters (European Environment Agency, 2020). The database is compiled from data sent by the national European water agencies involved in the Water Framework Directive (WFD).

Over 600 water quality parameters are included in the full dataset out of which 15 matched those of GRQA (Table 1). Out of all source datasets, WATERBASE had the shortest time series with observations covering only the period 2008–2018. The maximum site count per parameter is 1,976, while there were on average only around 19 observations per site.

In May 2020, the ICWRGC announced that parts of WATERBASE had been also added to the GEMSTAT database (International Centre for Water Resources and Global Change, 2020). However, only sites with more than three years of data were included in this update. As mean time series length per site was only 1.4 years in WATERBASE, a significant number of sites were left out, which is why we decided to include WATERBASE separately in GRQA. Although it is likely that there were many observations, which appeared both in GEMSTAT and WATERBASE, the duplication detection procedure discussed in section 3.3 should have identified them.

## 2.5 WQP

USGS, the U.S. Environmental Protection Agency (EPA) and the National Water Quality Monitoring Council developed the Water Quality Portal (WQP), which is so far the largest standardised water quality database (Read et al., 2017; United States Geological Survey, 2020). Although the portal also includes data from a few other countries (e.g. Mexico, Pacific islands) associated with the National Water Information System (NWIS) network, only a very limited amount of non-US samples were available. For this reason, only US national data was selected to be added to GRQA.

Due to the size of the source dataset, the full set of parameters could not be downloaded at once. Therefore, a scripted download procedure was used to retrieve water quality samples and their corresponding sampling sites separately per parameter. In the case of temperature, the data had to be further divided by state. Unlike other source datasets used in the study, the WQP often had multiple versions of the same parameter available under separate codes, in case the parameter had been measured in different units, using different methods, etc. The final count of parameters used for GRQA was 37 (Table 1).

The longest time series of source datasets is present in the WQP with some dating back to 1898. However, the average time series length per station is just over three years. Like GEMSTAT, WQP is still being updated, so most parameters have their

latest observations from 2020. Site count ranges from a single station (dissolved inorganic nitrogen) to 59,000 per parameter (total suspended solids).

## 3 Methodology

The GRQA compilation workflow was divided into three parts: (1) The pre-processing stage involved converting observation data from the five sources into a common format and harmonizing the corresponding metadata; (2) Pre-processed data were merged by parameter, after which outliers and time series characteristics were detected; (3) Duplicate detection was conducted in the last processing step. The Pandas (McKinney et al., 2010), GeoPandas (Jordahl et al., 2020) and NumPy (Harris et al., 2020) Python libraries were used throughout all data processing stages.

### 3.1 Source data preprocessing

*Parameter selection.* The parameters included in GRQA cover the four groups of water quality indicators outlined in the introduction: nutrients, carbon, sediments and oxygen (Table 7). GLORICH was used as a reference for parameter selection due to being one of the two global source datasets and having the least amount of discrepancies within source data, i.e. each GLORICH parameter had a single matching code, unit, etc.

*Parameter harmonization.* Preliminary analysis showed that there were ambiguities in the parameter names, codes, units and chemical forms in the different source datasets, which has been identified as a recurring issue when dealing with multi-source water quality data (McMillan et al., 2012; Sprague et al., 2017). For this reason, lookup tables were created for each of the source datasets (*\*_code_map.csv*) to use as guides in the following processing stages (Table 3). The purpose of the schemas was to match parameter codes and other metadata with the versions used later in the GRQA. For most parameters, this could be done based on the literal names, remarks and descriptions in the metadata. Relevant literature and online resources were consulted for more ambiguous scenarios. One such example was total suspended solids (TSS), which can also be reported as suspended particulate matter (SPM) (Neukermans et al., 2012). Where a reliable decision could not be made (e.g. biological oxygen demand as BOD vs BOD5) the parameters were kept separate.

*Unit conversion.* Units of measurement were harmonized along with other metadata. All parameters except temperature (°C), pH and dissolved oxygen (%) were converted into mg/l, which was the most prevalent unit in source data. Where units were converted, observation values had to be changed as well. This was done by calculating conversion constants, which were based on both the magnitude of the source unit (e.g. $\mu$g/l vs mg/l) and the reported chemical form of the parameter. The latter affected nitrite ($NO_2$), nitrate ($NO_3$) and ammonium ($NH_4$) the most, as these parameters had a variety of forms in the source data that were all converted into corresponding nitrogen versions ($NO_2$-N, $NO_3$-N & $NH_4$-N). In some cases, the chemical form could be identified from the source unit (e.g. mg{N}/L or mg{$NO_3$}/L), while others were detected by examining parameter names and method descriptions (e.g. "Nitrate, reported as nitrogen"). Where possible, additional information about these missing forms was collected from proxy sources, such as other similar datasets (e.g. Börker et al. (2020) in the case of GLORICH). These references have been included in the *form_ref* column in corresponding lookup tables (*\*_code_map.csv*). For other nitrogen

**Table 3.** Summary table of lookup table attributes.

| Attribute name | Description | Data type |
|---|---|---|
| source_param_code | Parameter code in source dataset | string |
| source_param_code_meta | Additional code specification used for CESI | string |
| param_code | Parameter code in GRQA | string |
| source_param_name | Parameter name in source dataset | string |
| param_name | Parameter name in GRQA | string |
| source_param_form | Parameter chemical form in source dataset | string |
| param_form | Parameter chemical form in GRQA | string |
| form_ref | Parameter form reference | string |
| source_unit | Parameter unit in source dataset | string |
| divisor | Divisor applied to the observation value | float |
| multiplier | Multiplier applied to the observation value | float |
| conversion_constant | Unit conversion constant calculated based on divisor and multiplier and applied to the observation value | float |
| unit | Parameter unit in GRQA | string |
| source | Source dataset name | string |

(TKN, TN, etc.), all carbon (DOC, TC, etc.) and phosphorus (TP, TIP, etc.) parameters, the chemical were assumed to be either N, C or P even if not reported, because there is only one common element in the molecule (Sprague et al., 2017). GLORICH was the only source dataset, which also needed conversion constants for carbon and phosphorus parameters as they had been reported as $\mu$mol/l. All WQP units matched those intended to be used for GRQA, so no conversion was needed. The formula for conversion constants was

$$x_2 = \frac{x_1 \times M_{x_2}}{n \times M_{x_1}} \tag{1}$$

where $x_1$ and $x_2$ are observation values before and after conversion, $M$ is the corresponding molar mass and $n$ the magnitude difference between source and converted unit. Some examples of unit conversion are given in Table 4. The full list of all unit conversion procedures is given in the appendix (Table A1).

*Site ID duplication.* There were some instances of duplicated site IDs in GLORICH (2 site pairs) and WATERBASE (101 pairs) source data, which meant that joining observations with sites would have created duplicate time series as well. Site ID duplicates could indicate that were have been small shifts in the site location or that the site had been closed and reinstated at some point. If the distance between the duplicate pairs was less than a kilometer, only the first instance was retained in the output table. When distance was greater than a kilometer both instances were removed as metadata that could be used to make a decision (e.g. when the site first opened) was not available. Finally, all duplicate pairs were exported as separate files (e.g. *GLORICH_dup_sites*).

**Table 4.** Examples of unit conversion from the chemical form in source data to the GRQA version. $x_1$ and $x_2$ are observation values before and after conversion, respectively.

| Parameter code | Source | Form | Source form | Unit | Source unit | $x_1$ | $M_{x_2}$ | $n$ | $M_{x_1}$ | $x_2$ |
|----------------|--------|------|-------------|------|-------------|-------|-----------|-----|-----------|-------|
| TAN | CESI | N | NH3 | mg/l | mg/l | 0.106 | 14.007 | 1 | 17.031 | 0.087 |
| NO2N | GEMSTAT | N | NO2 | mg/l | mg/l NO2 | 0.024 | 14.007 | 1 | 46.005 | 0.007 |
| NO3N | GLORICH | N | NO3 | mg/l | $\mu$mol/l | 210.268 | 14.007 | 1000 | 62.004 | 0.048 |
| NH4N | WATERBASE | N | NH4 | mg/l | mg/l | 0.063 | 14.007 | 1 | 18.039 | 0.049 |

*Coordinate conversion.* CESI and WQP originally had the site coordinates in the North American Datum of 1983 (NAD83). The Pyproj (Snow et al., 2020) Python library was used for converting the North American site coordinates into World Geodetic System 1984 (WGS84) which was the coordinate system chosen for the GRQA.

*Observation data filtering.* Preliminary cleaning included the removal of observations of negative, missing or low quality values. In this case, low quality refers to measurements that were flagged as either coming from unreliable sources or having any kind of literal quality assessment flag in the source data (e.g. "poor quality"). Additionally, observations marked as below (<) or above (>) detection limit in source data where flagged as such in GRQA as well (column *detection_limit_flag*). Observations originating from unreliable sources or otherwise suspect (e.g. unvalidated) were omitted. Three source datasets (GEMSTAT, 190  GLORICH & WATERBASE) had this type of a quality evaluation included in the metadata. Observations from sites marked as "Not for publication" due to national legislation in WATERBASE were also not included in GRQA.

   *Filtration information.* Where possible, supplementary information about whether a sample was filtered or unfiltered was retained as filtration can affect the sample values (Sprague et al., 2017). This information was usually available in a separate metadata column. Both "filtered" and "dissolved" were used depending on the source. GRQA includes the dissolved versions of 195  certain parameters (total nitrogen, total phosphorus and Kjeldahl nitrogen), which originally did not exist as separate parameters in WATERBASE and WQP. In those cases, the filtered/dissolved observations of TN, TP and TKN in the two datasets were treated as the corresponding dissolved forms (TDN, TDP, DKN) in GRQA.

   *Time and date processing.* Observations could have invalid timestamps due to formatting or entry errors, so a validity check was included in the pre-processing scripts. Dates were tested against the presumed source format and observations with incor- 200  rectly formatted or implausible dates were removed. The source datasets used different date formats, which were all converted into a common one (%Y-%m-%d). Were possible, observation time was extracted as well. A default value (00:00:00) was used to fill missing information. Time zone information was only possible to extract from the WQP. Other sources lacked time zone information, so it was not possible to determine whether the recorded timestamp was in local or Coordinated Universal Time (UTC) and the time given is up to the user to interpret.

*Other metadata.* If available, metadata about the upstream basin area, its unit and the name of the greater drainage region of the site was included in GRQA. Additional information about methods used or other available observation remarks in the source data were also retained. The metadata depended on the source and was available only sporadically and could not be concatenated in a reasonable way between the datasets, so the information is given in the GRQA for each source separately in

the format of *source_meta_sourcecolumnname* (e.g. *GEMSTAT_meta_Analysis Method Code*). Here, the source column names were kept as they appear in raw data, e.g. spaces were not replaced with underscores.

## 3.2 Outlier treatment, time series availability and continuity

*Time series availability and continuity.* The analysis of the statistics generated during pre-processing showed that most of the time series extracted from the source datasets are very discontinuous. For example, the mean time series length per site for total phosphorus (TP) in GEMSTAT was 6.6 years and 4.9 years in GLORICH, while the mean observation count per site was only 57.7 and 52.4, respectively. This means that many sites have observations at a monthly time step at best. Similar findings have been previously reported about WQP time series (Read et al., 2017; Shen et al., 2020).

In order to illustrate the suspected temporal fragmentation in observation data, monthly availability and monthly continuity statistics appropriated from the strategy used by Crochemore et al. (2019) were calculated for each site in each of the merged parameter time series. Both characteristics can give insight to the granularity of the time series and can affect the applicability of different modeling methods. Monthly availability of observation data was defined as the ratio between number of months with at least one observation and the total number of months a particular site had any observations. A ratio of 1.0 would mean that there was at least one observation in every month of the time series. Monthly continuity was calculated as the ratio between the longest period of consecutive months with any measurements and the length of time series in months. Here, a ratio of 1.0 would mean that there were no months without observations and the time series is continuous on a monthly level. The resulting characteristics were added as columns in the output files.

*Outlier flagging.* Water quality modeling often involves dealing with numerous outliers and uncertainties in observation data, particularly when integrating time series from multiple sources (McMillan et al., 2012; Sprague et al., 2017). Due to the differences in environmental conditions and water regimes, the potential range of observation values can vary a lot between catchments. Although extreme outliers caused by faulty equipment or data entry errors can sometimes be detectable by examining distribution plots, it is often difficult to decide whether an outlier is an error or not. For example, sudden spikes in observation time series can be caused by events such as accidental fertilizer spills to the waterway or a cow getting entrapped in a in-stream wetland (Hughes et al., 2016), which can have short-term effects on water quality and, therefore, should not be removed from data. However, flagging outliers can still help researchers troubleshoot potential issues at the modeling stage.

For this reason, no observations were omitted from the time series and two flags associated with outliers were added to the output tables instead. First flag (*obs_iqr_outlier*) shows whether an observation was deemed to be an outlier by the interquartile range ($IQR$) test. $IQR$ is defined as the difference between the third ($Q3$) and first ($Q1$) quartile. All values greater than $Q3 + 1.5 \times IQR$ or less than $Q1 - 1.5 \times IQR$ are considered outliers. The second flag (*obs_percentile*) was an indicator (0.0–1.0) showing which percentile a particular observation belongs to. Histograms along with box and whisker plots were used to visually show the range and distribution of the parameter observations. The plots were produced for every parameter and are included in the GRQA data repository.

### 3.3 Duplicate observation detection

The global datasets (GEMSTAT and GLORICH) used in this study had at least partial spatial overlap with the other three sources, which means that merging could have created duplicate sites in the GRQA. Contrary to site ID duplicates within the same dataset discussed in section 3.1, site duplicates from different sources would likely also have different IDs. Therefore, rather than comparing ID information, the duplicates had to be identified by spatial proximity and time series similarity. Similar to procedures described in section 3.2, duplicate detection was done separately for each parameter.

First stage of duplicate detection was clustering sites based on their geographic location. The DBSCAN (density-based spatial clustering of applications with noise) algorithm (Xu et al., 1998) from the Scikit-learn Python library (Pedregosa et al., 2011) was used to create clusters of sites within a one kilometer radius of each other, which is the approximate accuracy of around two decimal points in latitude/longitude degrees. There does not seem to be a consensus for assigning this search radius for duplicate detection and the assessment of spatial proximity depends on the subjective threshold set by authors. For example, the GSIM streamflow dataset (Do et al., 2018) used a radius of 5 km for selecting potential duplicate gauging stations. The 1 km radius was chosen to avoid having too many false positives (e.g. in the case of small headwater catchments) to evaluate in the second stage of deduplication (RMSE calculation). A major advantage of DBSCAN compared to similar density-based clustering methods is that the algorithm can be run without determining a priori the number of output clusters (Birant and Kut, 2007). In addition, DBSCAN has shown to be more applicable than others when dealing with large-scale datasets (Khan et al., 2014; Parimala et al., 2011).

Although there are time series similarity detection methods that can be applied to irregular time series and handle some degree of discontinuity, the focus of those methods is on misalignment of the time of observations rather than differences in the pattern of time series gaps (Berndt and Clifford, 1994). Therefore, it is likely that GRQA time series are too fragmented for these advanced methods to yield reliable results. A conservative approach based on root-mean-square error (RMSE) was chosen here instead. Output site clusters were converted into unique site pairs, so that all sites within a cluster could be compared to one another (e.g. a cluster of four would yield six unique ID pairs). Site ID pairs were then used to extract corresponding time series from observation data. Only observations made on matching dates were used for calculating the RMSE and only pairs where RMSE was equal to zero were considered as potential duplicates. Finally, the duplicates were exported into separate CSV files (e.g. *TP_dup_obs.csv*) along with relevant metadata to help the user decide whether the sites can be considered duplicate (Table 5). A high number of matching dates with the same observation value (column *date_match_count*) would indicate a higher likelihood of duplication.

## 4 Results

*GRQA data model and descriptive overview.* The GRQA dataset consists of observation time series for 42 different water quality parameters provided in tabular form as CSV files. Each of the observation files is accompanied by corresponding metadata files (tables and images) describing the spatial and temporal characteristics of the time series.

GRQA is made up of the following files (Fig. 1):

**Table 5.** Summary table of duplicate observation file attributes.

| Attribute name | Description | Data type |
|---|---|---|
| obs_id_1 | Observation ID of first site | string |
| lat_wgs84_1 | Latitude of first site | float |
| lon_wgs84_1 | Longitude of first site | float |
| site_id_1 | First site ID | string |
| site_name_1 | First site name | string |
| obs_value_1 | First site observation value | float |
| source_1 | First site source | string |
| site_ts_availability_1 | First site availability | float |
| site_ts_continuity_1 | First site continuity | float |
| obs_date | Observation date | string |
| obs_id_2 | Observation ID of second site | string |
| lat_wgs84_2 | Latitude of second site | float |
| lon_wgs84_2 | Longitude of second site | float |
| site_id_2 | Second site ID | string |
| site_name_2 | Second site name | string |
| obs_value_2 | Second site observation value | float |
| source_2 | Second site source | string |
| site_ts_availability_2 | Second site availability | float |
| site_ts_continuity_2 | Second site continuity | float |
| date_match_count | Number of matching dates with the same observation value | int |
| param_code | Parameter code | string |

– A data catalog (*GRQA_data_catalog.pdf*) with maps showing the spatiotemporal coverage and graphs describing the distribution of all 42 parameters along with a README file describing the dataset structure

– Water quality observation time series files (named *paramcode_GRQA.csv*)

– GRQA metadata (folder *meta*) containing descriptive statistics (*GRQA_param_stats.csv*) and duplicate observation files (*source_dup_obs.csv*), where relevant

– The set of overview figures (folder *figures*) contains

Histograms (*paramcode_GRQA_hist.png*) and box plots (*paramcode_GRQA_box.png*) showing the distribution of observation values by source dataset

Maps showing the spatial distribution of the observations by source (*paramcode_GRQA_spatial_dist.png*)

Maps showing the median observation values of sites (*paramcode_GRQA_median.png*)

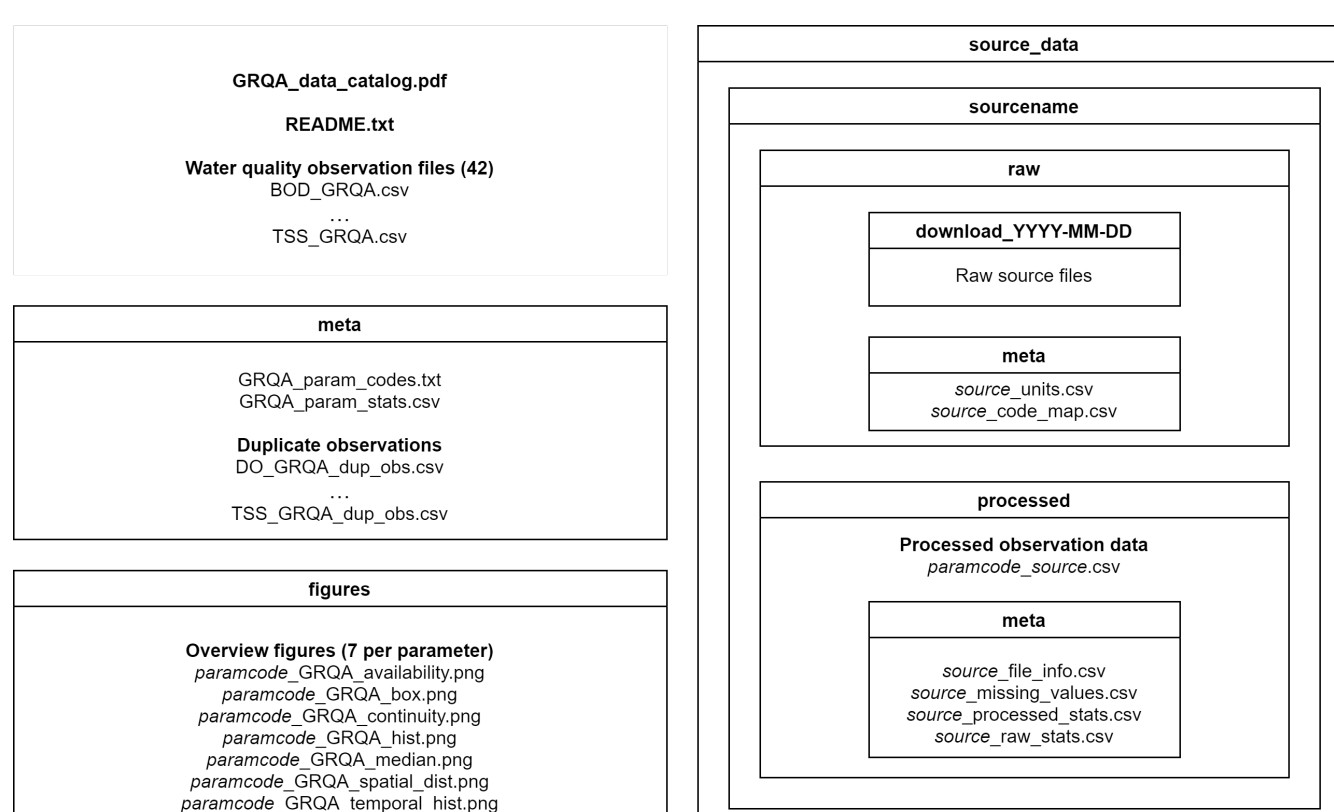

**Figure 1.** Diagram showing the folder structure and contents of the GRQA dataset.

Maps showing the monthly availability (*paramcode_GRQA_availability.png*) and continuity (*paramcode_GRQA_continuity.png*) of the observations

The five source datasets are also included in the GRQA data package. Folder *source_data* includes

– The *raw* folder with downloaded source files and harmonization schemas used in the preprocessing stage (*source_code_map.csv*) for each source dataset along with the original units (*source_units.csv*)

– The *sourcename/processed* folder contains summary statistics of observation values by parameter for each source dataset before (*paramcode_source_raw_stats.csv*) and after (*paramcode_source_processed_stats.csv*) processing along with information about the number of missing values (*source_missing_values.csv*) and source file size (*source_file_info.csv*)

– Where relevant, *processed/meta* also includes duplicate site ID files (*source_dup_sites.csv*)

**Table 6.** Summary table of output water quality observation file attributes.

| Attribute name | Description | Data type |
|---|---|---|
| obs_id | Unique observation ID generated by hashing | string |
| lat_wgs84 | Observation site latitude in WGS84 | float |
| lon_wgs84 | Observation site longitude in WGS84 | float |
| obs_date | Observation date in the %Y-%m-%d format | string |
| obs_time | Observation time in the %H:%M:%S format | string |
| obs_time_zone | Observation time zone code | string |
| site_id | Observation site ID | string |
| site_name | Observation site name | string |
| site_country | Observation site country | string |
| upstream_basin_area | Site upstream basin area | string |
| upstream_basin_area_unit | Site upstream basin area unit | string |
| drainage_region_name | Drainage region where site is located in | string |
| param_code | Parameter code in GRQA | string |
| source_param_code | Parameter code in source dataset | string |
| param_name | Parameter name in GRQA | string |
| source_param_name | Parameter name in source dataset | string |
| obs_value | Observation value in GRQA | float |
| source_obs_value | Observation value in source dataset | float |
| detection_limit_flag | Whether a value was flagged as below (<) or above (>) detection limit in source data | string |
| param_form | Parameter chemical form in GRQA | string |
| source_param_form | Parameter chemical form in source dataset | string |
| unit | Parameter unit in GRQA | string |
| source_unit | Parameter unit in source dataset | string |
| filtration | Sample filtration information | string |
| source | Source dataset name | string |
| obs_percentile | Percentile of the observation value | float |
| obs_iqr_outlier | Flag to mark whether observation value is an outlier according to the interquartile range test | string |
| site_ts_availability | Monthly availability of the time series per site | float |
| site_ts_continuity | Monthly continuity of the time series per site | float |
| *_meta_* | Other observation metadata with a reference to the corresponding source column (e.g., GEMSTAT_meta_Method Description) | string |
| … | … | |

**Table 7.** GRQA water quality parameter statistics.

| Parameter code | Parameter name | Sites | Observations | Median value | Unit | Start year | End year | Outlier % |
|---|---|---|---|---|---|---|---|---|
| BOD | Biochemical Oxygen Demand | 2,945 | 163,531 | 2.627 | mg/l | 1974 | 2019 | 13.4 |
| BOD5 | Biochemical Oxygen Demand (BOD5) | 13,283 | 278,629 | 5.875 | mg/l | 1905 | 2020 | 8.3 |
| BOD7 | Biochemical Oxygen Demand (BOD7) | 386 | 5,282 | 2.200 | mg/l | 2013 | 2018 | 5.9 |
| COD | Chemical Oxygen Demand | 2,769 | 126,372 | 22.362 | mg/l | 1974 | 2019 | 10.8 |
| CODCr | Chemical Oxygen Demand (Cr) | 671 | 7,350 | 24.900 | mg/l | 2013 | 2018 | 3.4 |
| CODMn | Chemical Oxygen Demand (Mn) | 287 | 2,310 | 4.600 | mg/l | 2013 | 2018 | 2.3 |
| DC | Total Dissolved Carbon | 7 | 9 | 4.800 | mg/l | 2000 | 2001 | 0 |
| DIC | Dissolved Inorganic Carbon | 969 | 30,633 | 12.266 | mg/l | 1968 | 2020 | 3.5 |
| DIN | Dissolved Inorganic Nitrogen | 119 | 7,822 | 4.200 | mg/l | 1998 | 2019 | 2.6 |
| DIP | Dissolved Inorganic Phosphorus | 9,931 | 612,922 | 0.040 | mg/l | 1942 | 2017 | 13.3 |
| DKN | Dissolved Kjeldahl Nitrogen | 2,820 | 80,732 | 0.347 | mg/l | 1973 | 2020 | 6.5 |
| DO | Dissolved Oxygen | 48,072 | 1,487,724 | 8.835 | mg/l | 1898 | 2020 | 2.2 |
| DOC | Dissolved Organic Carbon | 14,799 | 413,328 | 2.804 | mg/l | 1968 | 2020 | 6.8 |
| DON | Dissolved Organic Nitrogen | 10,811 | 163,630 | 0.371 | mg/l | 1951 | 2020 | 8.1 |
| DOP | Dissolved Organic Phosphorus | 142 | 899 | 0.010 | mg/l | 1971 | 2003 | 8.7 |
| DOSAT | Dissolved Oxygen Saturation | 34,949 | 953,274 | 92.164 | % | 1898 | 2020 | 8.7 |
| NH4N | Ammonium Nitrogen | 11,372 | 651,850 | 0.027 | mg/l | 1942 | 2018 | 15.1 |
| NO2N | Nitrite Nitrogen | 30,902 | 720,944 | 0.010 | mg/l | 1900 | 2020 | 12.7 |
| NO3N | Nitrate Nitrogen | 45,422 | 1,229,584 | 0.468 | mg/l | 1900 | 2020 | 11.1 |
| PC | Particulate Carbon | 2,898 | 51,049 | 0.908 | mg/l | 1995 | 2020 | 11 |
| pH | pH | 27,577 | 1,372,794 | 6.886 | pH | 1900 | 2020 | 14.1 |
| PIC | Particulate Inorganic Carbon | 1,095 | 9,196 | 0.060 | mg/l | 1974 | 2020 | 14 |
| PN | Particulate Nitrogen | 2,996 | 56,125 | 0.129 | mg/l | 1981 | 2020 | 9.5 |
| POC | Particulate Organic Carbon | 22,910 | 615,941 | 1.617 | mg/l | 1900 | 2020 | 9.7 |
| PON | Particulate Organic Nitrogen | 28 | 1,111 | 0.120 | mg/l | 1989 | 2019 | 14 |
| POP | Particulate Organic Phosphorus | 12 | 13 | 0.020 | mg/l | 1999 | 2000 | 7.7 |
| TAN | Total Ammonia Nitrogen | 27,980 | 717,776 | 0.065 | mg/l | 1900 | 2020 | 13.3 |
| TC | Total Carbon | 1,181 | 12,338 | 27.000 | mg/l | 1968 | 2007 | 3.3 |
| TDN | Total Dissolved Nitrogen | 968 | 62,980 | 0.310 | mg/l | 1972 | 2020 | 11.2 |

| Parameter code | Parameter name | Sites | Observations | Median value | Unit | Start year | End year | Outlier % |
|---|---|---|---|---|---|---|---|---|
| TDP | Total Dissolved Phosphorus | 3,325 | 169,297 | 0.031 | mg/l | 1965 | 2020 | 11.3 |
| TEMP | Water Temperature | 26,860 | 1,113,471 | 18.968 | Deg C | 1912 | 2020 | 9.3 |
| TIC | Total Inorganic Carbon | 1,984 | 23,024 | 11.833 | mg/l | 1968 | 2019 | 3.8 |
| TIN | Total Inorganic Nitrogen | 78 | 12,951 | 3.649 | mg/l | 1992 | 2020 | 0.8 |
| TIP | Total Inorganic Phosphorus | 1,328 | 42,495 | 0.026 | mg/l | 1971 | 2018 | 13.8 |
| TKN | Total Kjeldahl Nitrogen | 9,418 | 425,595 | 0.680 | mg/l | 1962 | 2020 | 8.1 |
| TN | Total Nitrogen | 18,507 | 575,887 | 1.329 | mg/l | 1958 | 2020 | 11.9 |
| TOC | Total Organic Carbon | 18,032 | 420,029 | 4.526 | mg/l | 1958 | 2020 | 7.2 |
| TON | Total Organic Nitrogen | 22,799 | 592,654 | 0.622 | mg/l | 1900 | 2020 | 8.6 |
| TOP | Total Organic Phosphorus | 294 | 1,811 | 0.030 | mg/l | 1971 | 2020 | 11.9 |
| TP | Total Phosphorus | 44,990 | 1,914,538 | 0.105 | mg/l | 1900 | 2020 | 11.8 |
| TPP | Total Particulate Phosphorus | 77 | 5,836 | 0.021 | mg/l | 1978 | 2019 | 10.5 |
| TSS | Total Suspended Solids | 68,592 | 1,958,429 | 9.785 | mg/l | 1898 | 2020 | 20.5 |

The structure of GRQA observation files is given in Table 6. In addition to the attributes outlined in section 3, the extracted metadata also includes information about the upstream basin and drainage region of the observation site. It has to be noted that the availability of this information was dependent on both the source (i.e. not present in CESI and WATERBASE) and the observation site itself and is therefore available only sporadically in GRQA as well (Table 6). Parameter codes, names, forms and observation values in GRQA are given as they appeared in source data alongside their harmonized and processed GRQA versions, so that end users could assess the validity of conversion and make corrections if needed.

Statistical overview of the parameters included in GRQA is shown in Table 7. The number of sites per parameter ranges from only 7 (DC) up to 68,592 (TSS). Parameters having more sites generally also have more observations. Parameters with a small number of sites and observations were usually present in only one or two source datasets. For example, dissolved organic phosphorus (DOP) only existed in WQP. Different versions of biochemical and chemical oxygen demand that could not be harmonized based on source metadata were kept separate, although the median value for BOD and BOD5 ended up being equal.

Spatial distribution of water quality observation sites depended on the parameter and is illustrated in Fig. 2 using dissolved oxygen (DO), dissolved organic carbon (DOC), TP and TSS. These parameters were the largest in terms of number of sites and observations in their corresponding groups (oxygen, carbon, nutrients and sediments). They are also used in the following figures. Some observations that could be made when examining site maps were the following:

– Europe and North America are the best represented in the case of all parameters

– Coverage is also good in Australia, New Zealand, parts of East Asia and Brazil in the case of some of the key parameters (e.g. TP, TN)

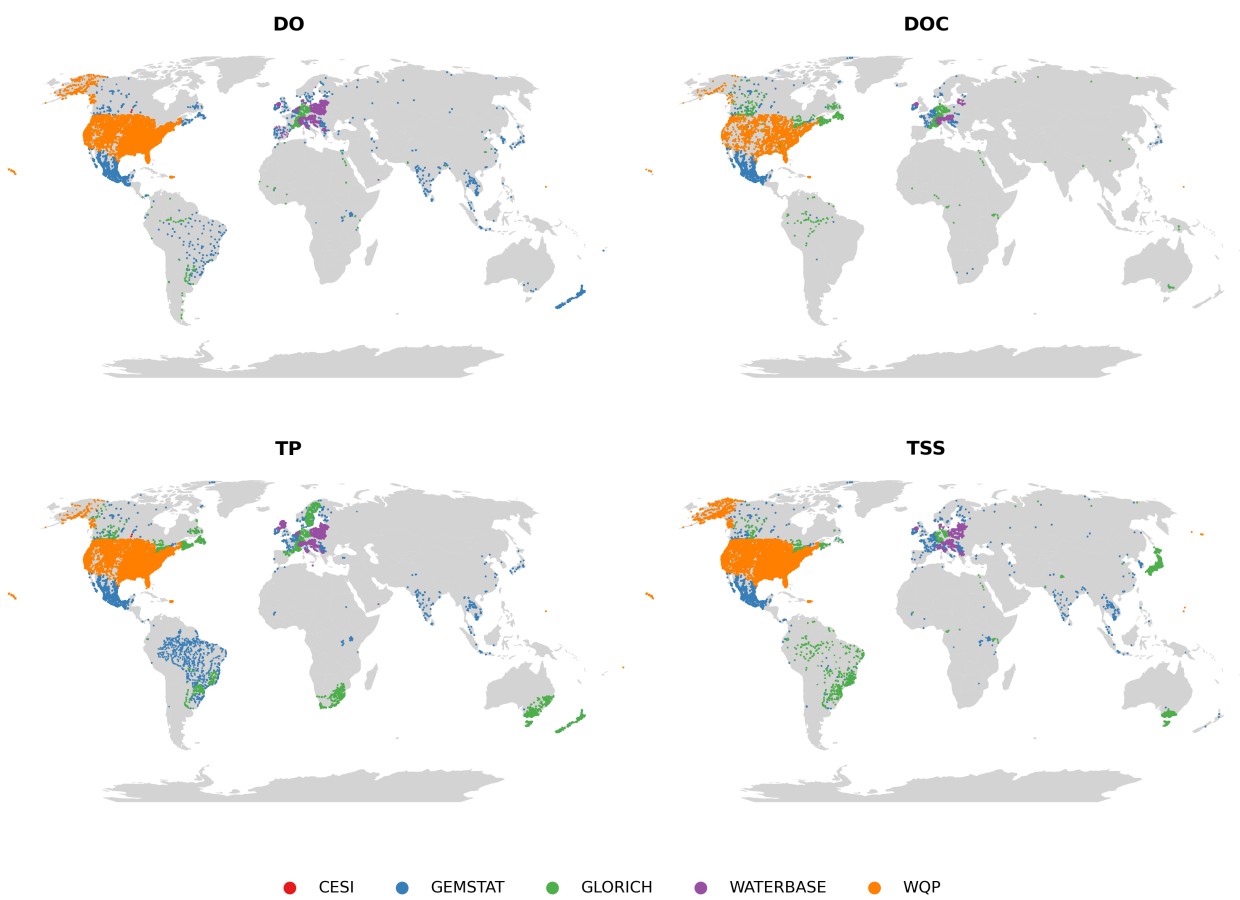

**Figure 2.** Distribution of observation sites for dissolved oxygen (DO), dissolved organic carbon (DOC), total phosphorus (TP) and total suspended solids (TSS).

– Rest of the world (Africa, most of Asia) only has sporadic coverage

The temporal distribution of the four parameters is given in Fig. 3. Similar to the spatial distribution, temporal coverage of observations depended on both source data and parameter with WQP having the longest and WATERBASE the shortest time series. Most of the data from GEMSTAT are from the past decade, while GLORICH has a more even observation distribution throughout the time series.

*Statistical characteristics of GRQA observation time series.* As mentioned in the previous section, each of the observation files was accompanied by a set of images and tables giving insight into the characteristics of the observation time series. The structure of tabular summary statistics is shown in Table 8. These files contain some basic statistics (standard deviation, etc) about observation values per parameter and source. In addition, information about the temporal characteristics of time series (mean length per site, etc) is given as well as this can be important when assessing the suitability of the data for modeling purposes.

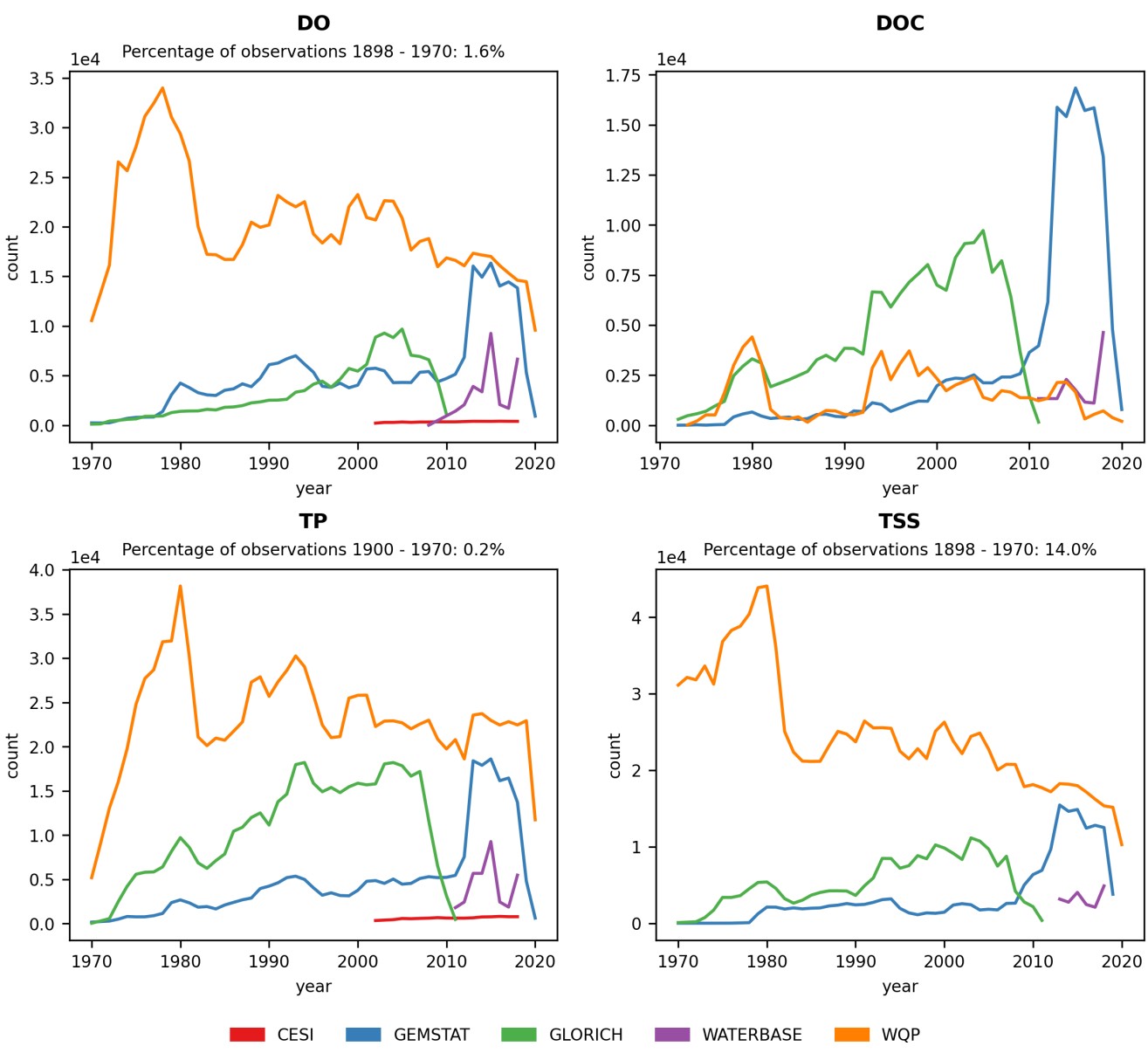

**Figure 3.** Temporal distribution of observations for dissolved oxygen (DO), dissolved organic carbon (DOC), total phosphorus (TP) and total suspended solids (TSS) for the period 1970–2020. Percentage of observations before the period shown on the plot is given for each parameter. Only seven observations ($1.69 \times 10^{-5}\%$) existed for DOC in the 1968–1970 period.

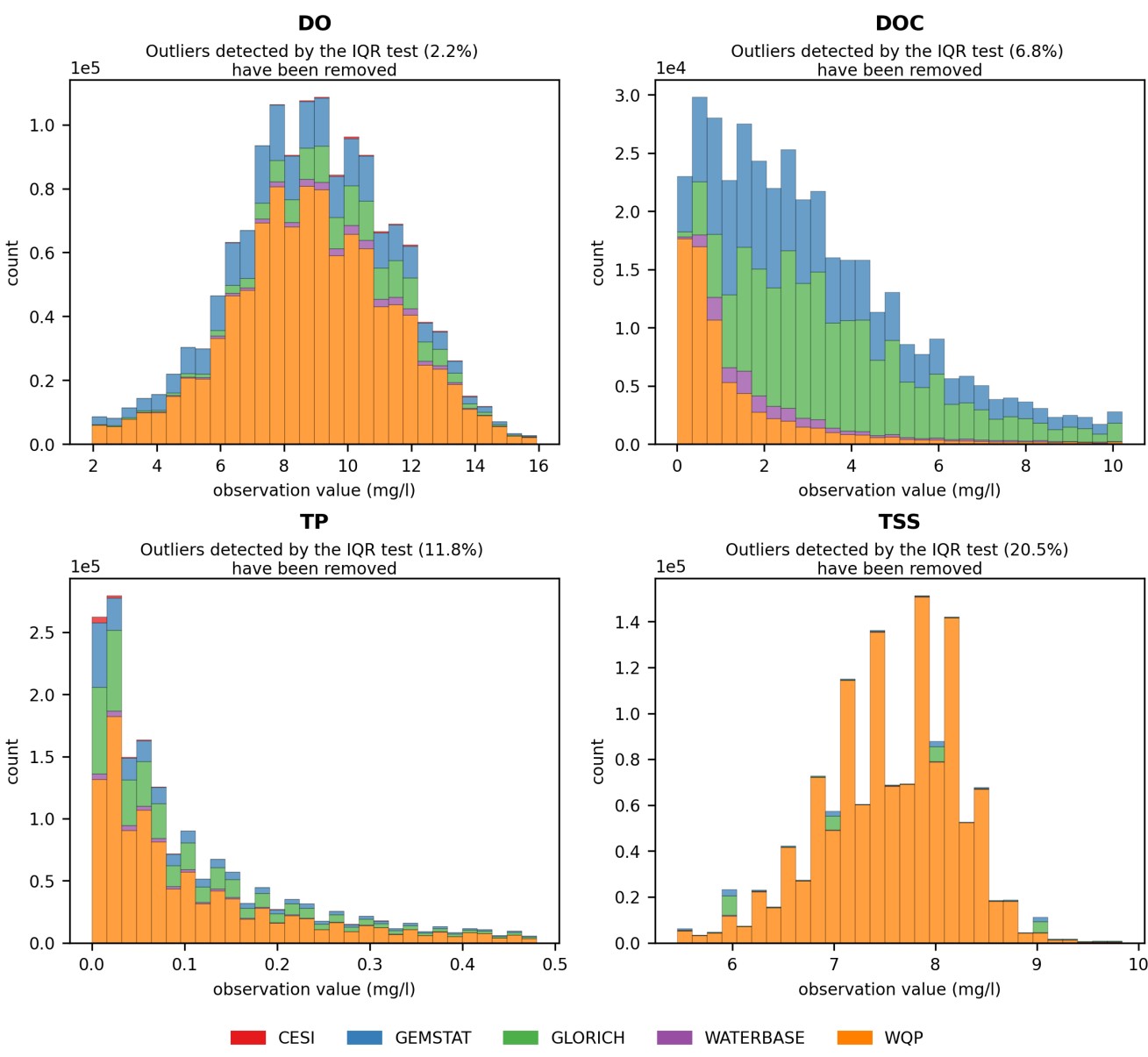

**Figure 4.** Distribution of observation values for dissolved oxygen (DO), dissolved organic carbon (DOC), total phosphorus (TP) and total suspended solids (TSS). Outliers determined by the IQR test (Table 7) are not shown on the plot.

**Table 8.** Summary table of observation time series statistics file attributes.

| Attribute name | Description | Data type |
|---|---|---|
| source_param_code | Parameter code in source dataset | string |
| param_code | Parameter code in GRQA | string |
| param_name | Parameter name in source dataset | string |
| source_param_form | Parameter form in source dataset | string |
| param_form | Parameter form in GRQA | string |
| source_unit | Parameter unit in source dataset | string |
| unit | Parameter unit in GRQA | string |
| count | Total number of observations | int |
| min | Minimum observation value | float |
| max | Maximum observation value | float |
| mean | Mean observation value | float |
| median | Median observation value | float |
| std | Standard deviation of observation values | float |
| min_year | Time series start | int |
| max_year | Time series end | int |
| ts_length | Total time series length per parameter | float |
| site_count | Total number of sites per parameter | int |
| mean_obs_count_per_site | Mean observation count per site | float |
| mean_ts_length_per_site | Mean time series length in years per site | float |

The applicability of water quality modeling is greatly affected by the distribution of observation values as a majority of modeling methods require a near normal distribution. The skewness caused by extreme outliers is a common problem in hydrological modeling. The observations often follow a lognormal distribution, which means that the data often needs to be transformed and normalized in order to be usable (Helsel, 1987; Hirsch et al., 1982; Parmar and Bhardwaj, 2014). Similar behavior was also examined in GRQA, where values of most parameters showed a strong positive skew. This can be seen in histograms (Fig. 4) and box plots (Fig. A1). For illustrative purposes, values determined as outliers by the IQR test have been omitted from the figures. In the case of parameters such as TP and TSS, the skewness remains even after outlier omission. This is confirmed by the box plots, where the total range of the values greatly exceeds the median.

Availability (Fig. 5) and continuity (Fig. 6) plots were used to examine the temporal fragmentation of the time series. In general, observations from national sources (CESI and WQP) exhibited slightly higher availability and continuity than others, likely caused by more consistent data acquisition frameworks. No clear spatial pattern emerged from the analysis meaning that differences in both indicators exist at the site level even within the same country. Due to how the metrics were calculated, shorter time series outperformed longer ones. An example of this is TP in Brazil, where the examined high continuity correlated

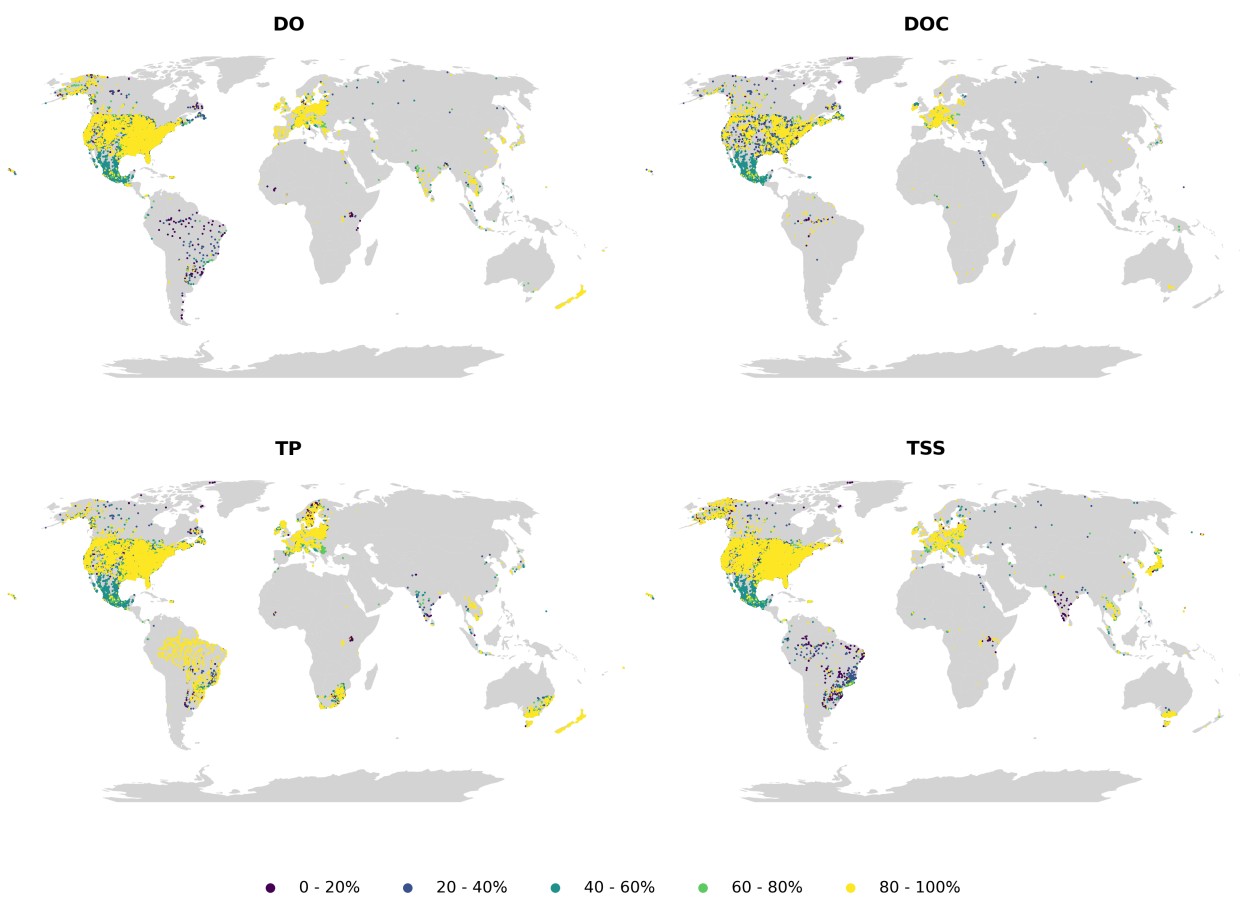

**Figure 5.** Monthly availability for dissolved oxygen (DO), dissolved organic carbon (DOC), total phosphorus (TP) and total suspended solids (TSS).

with very short mean time series length (less than a year). Parameters with very fragmented time series (e.g. TSS) had only a limited number of sites where observations had been collected consistently throughout the whole time frame.

The GRQA also includes plots of median observation values, which were calculated over the whole time series for each site. Seasonal fluctuations cannot be identified on this aggregation level, so the maps are meant to be only indicative. An example of median plots can be seen in the appendix (Fig. A2).

## 5   Discussion

### 5.1   Limitations and considerations regarding the use of GRQA

Taking into account aforementioned issues encountered during the compilation of GRQA, certain limitations and potential remaining errors have to be considered when using the dataset for water quality modeling.

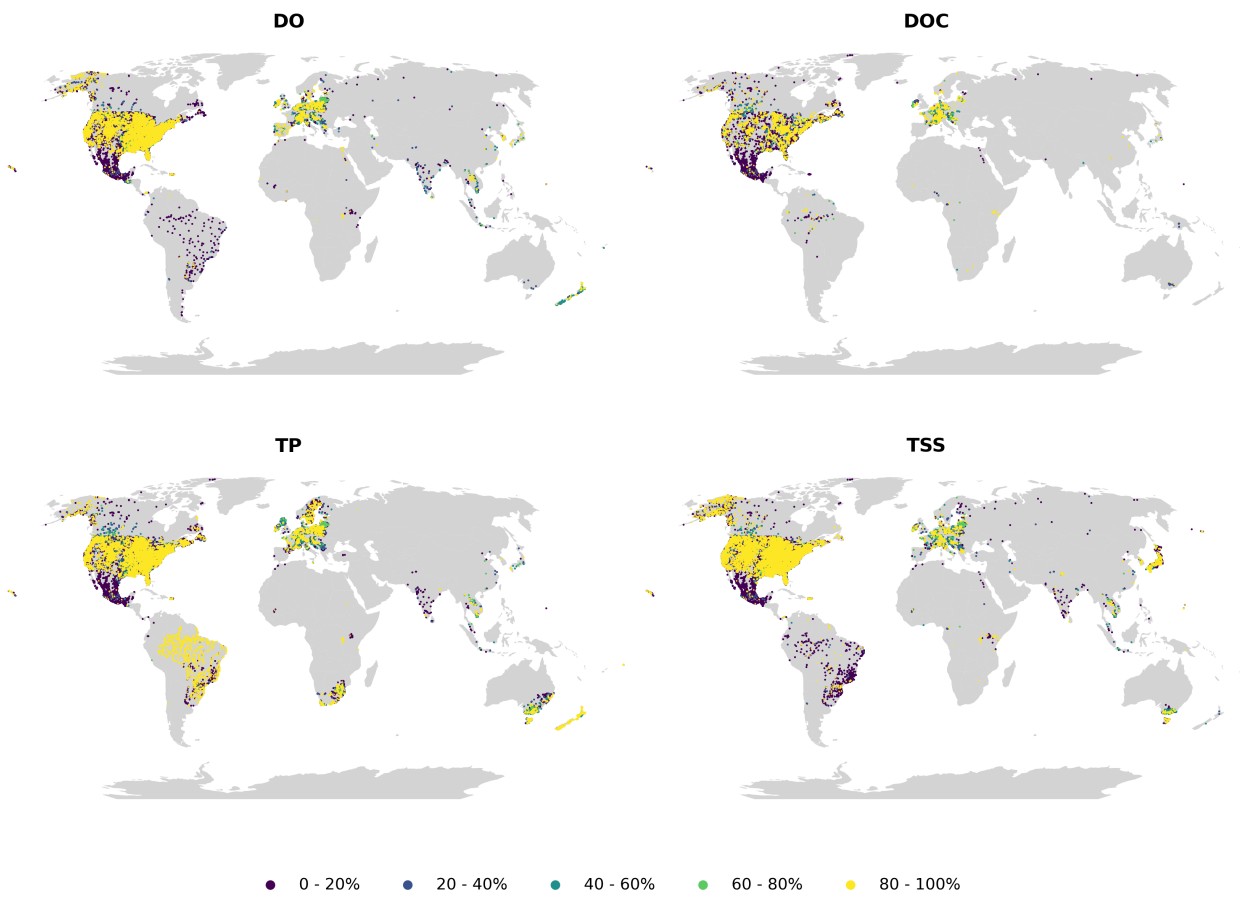

**Figure 6.** Monthly continuity for dissolved oxygen (DO), dissolved organic carbon (DOC), total phosphorus (TP) and total suspended solids (TSS).

*Potential errors in unit conversion.* As described in section 3, several assumptions had to be made when creating harmoniza-
tion schemas about the chemical form of certain nitrogen parameters ($NO_2$, $NO_3$ and $NH_4$). However, if the assumption made based on this limited ancillary information was incorrect then using the conversion would have been affected as well. For this reason, the source observation values along with source units were retained and the users can retrace the conversion steps using the harmonization schemas.

*Skewness of observation values.* The outlier treatment strategy used for GRQA involved only flagging the values based
on the IQR test, which means that the skewness illustrated in section 4 still remains. Although the described strong positive skew existed also in source data, potential unit conversion errors could have exaggerated it. As shown by histograms, omitting flagged outliers is not enough to eliminate the skewness in some cases (TP and TSS), so additional processing could be needed to transform the data into a normal shape. Power transformation methods like the Box-Cox transformation (Box and Cox, 1964) could be used to further minimize skewness. It is likely that some of the most extreme outliers are caused by data entry errors

or equipment malfunction rather than events such as agricultural spills. For setting thresholds to determine whether a value is illogical or not, more sophisticated outlier detection methods based on some general freshwater quality guidelines (Enderlein et al., 1996) could perhaps be used to further filter the observation values.

## 5.2 Suggestions for improving multi-source water quality data compilation

*Metadata quality*. When merging datasets from different sources, most of the complications stemmed from inadequate metadata
of water quality observations, such as ambiguous parameter names and codes, and missing details on the chemical forms of parameters. This information would be integral for harmonizing units and observation values. The terms used for indicating the filtration status of samples are often dependent on the interpretation of the authors (total vs unfiltered, dissolved vs filtered), which can affect results when merging (McMillan et al., 2012; Sprague et al., 2017). Annotation of suspect or incomplete data is another aspect of good quality metadata (Gudivada et al., 2017). Internal quality control measures such as the ones in
GEMSTAT and WATERBASE would help the end user in the data cleaning stage and eliminate some of the outliers.

The following aspects should be considered to make multi-source data harmonization more efficient in the future:

- Parameter forms should be reported with the units

- The filtration status of the samples should be reported and the terms filtered/unfiltered should be preferred as opposed to the more ambiguous dissolved/total

375 - Machine-readable quality flags as found in GEMSTAT (columns *Value Flags* and *Data Quality*) or WATERBASE (columns *resultObservationStatus*, *metadata_statusCode* and *metadata_observationStatus*) should be added

- Whether observations are daily or monthly at the source level should be clearly defined

- Area units (m$^2$, km$^2$, etc) should be included, when the upstream catchment area of the site is reported

- Other information about potential errors in the data (potential duplicates, typographical errors, etc)

- When certain assumptions or decisions are made when harmonizing data from different sources, they should be reported when the data is published

*Spatial and temporal discontinuity*. Although spatial coverage of water quality observations in GRQA exceeds that of the existing global datasets (GEMSTAT and GLORICH), large areas of Africa and Asia are empty. A major reason might be a lack of knowledge and funding to update and extend site networks, particularly in hard to reach areas. In addition, not all
385 governments adhere to an open data policy. Therefore, improving the spatial coverage of water quality data still relies mostly on implementing additional measures to encourage countries to share it in accordance with open data principles.

The availability and continuity analysis showed that the GRQA time series are fragmented and significant gaps remain in the data, which will negatively affect large-scale modeling performance. These gaps could be caused by both issues with sensor maintenance or technical limitations under certain conditions (weather, etc) and inconsistencies in the data acquisition

practices on the local level. Recently, ML based solutions for time series augmentation have been used to fill in gaps in historical monitoring data (Gao et al., 2018; Ren et al., 2019). However, this kind of gap filling still requires enough good quality training data in the existing time series fragments to be effective and can potentially only be of help when improving the temporal, rather than spatial coverage.

Another option for improving continuity is using data from one time series to fill in gaps in another. For example, turbidity has been successfully translated into TP and TSS content (Castrillo and García, 2020; Jones et al., 2011). As turbidity data can be acquired at a higher frequency than TP and TSS, the use of such surrogate parameters can be helpful in data scarce regions for certain parameters.

*General remarks.* An important part in improving the spatiotemporal coverage of water quality is raising awareness about the existing datasets (e.g. GEMSTAT), so that new institutions could join the contributor network and submit their own site data. Continued growth of international collaboration will be vital in improving open global water quality data (Blöschl et al., 2019; Tang et al., 2019). Most of the data collected locally is intended only for regional or national use. Thus, the data is not compatible with those from other countries due to lack of common metadata management practices with problems discussed above being a major bottleneck (Hutton et al., 2016; Sprague et al., 2017; Stagge et al., 2019). Providing those institutions with an example workflow when designing water quality data pipelines, such as the schema recently proposed by Plana et al. (2019), would help them develop their own data management strategy. The workflow used to compile GRQA along with the issues raised in this study will hopefully also help to draw attention to this topic.

## 6   Conclusions

The GRQA dataset was created with the intention to improve the spatiotemporal coverage of previously available open water quality data and provide an example workflow for multi-source data compilation that can be accustomed for other data sources as well. The current version of GRQA is mainly focused on different forms of the main nutrients (N and P) and carbon compounds, although GEMSTAT, WATERBASE and WQP also had many other types of parameters that are used as water quality indicators (heavy metals, pesticides, etc). Other researchers are able to make additions and customize the dataset to their needs for parameter-specific studies using the scripts published with GRQA.

Updates and additions by the hydrological community are encouraged to further develop GRQA. As it stands, GRQA is a set of well structured CSV files rather than a queryable database. We intend to add a Jupyter Notebook example of loading and processing the CSV files to the GRQA GitHub repository. We included an extensive data catalogue with graphs and maps for temporal and spatial coverage of every variable as supplementary material. This should help potential users to get a better overview of the data before downloading it. Converting the files into a database would also greatly improve data management and make extending GRQA easier in the future. In the case of a relational database, the schema recommended by Plana et al. (2019) could be followed. We also consider the addition of an online dashboard for data visualization and download, similar to that of GEMSTAT or WQP. A versioning system along with a metadata validation strategy similar to Welty et al. (2020) could be implemented to ensure metadata quality.

Future work could also include the development of a dataset for catchment characteristics in order to better study how water quality in rivers and streams is affected by land use changes in their catchments. The CAMELS dataset (Addor et al., 2017) and its regional implementations (Chagas et al., 2020; Coxon et al., 2020) can be used as an example. In addition, interactions between water quality and streamflow can be further studies by linking water quality observations to streamflow data from the Global Streamflow Indices and Metadata Archive (GSIM) (Do et al., 2018).

*Code and data availability.* The GRQA dataset, supplementary metadata and figures are available for download on the DataCite and OpenAire enabled Zenodo repository https://doi.org/10.5281/zenodo.5097436 (Virro et al., 2021).

The data processing scripts used for the compilation of GRQA are available on Zenodo https://doi.org/10.5281/zenodo.5082147 (Virro and Kmoch, 2021).

## Appendix A: Figures and tables in appendices

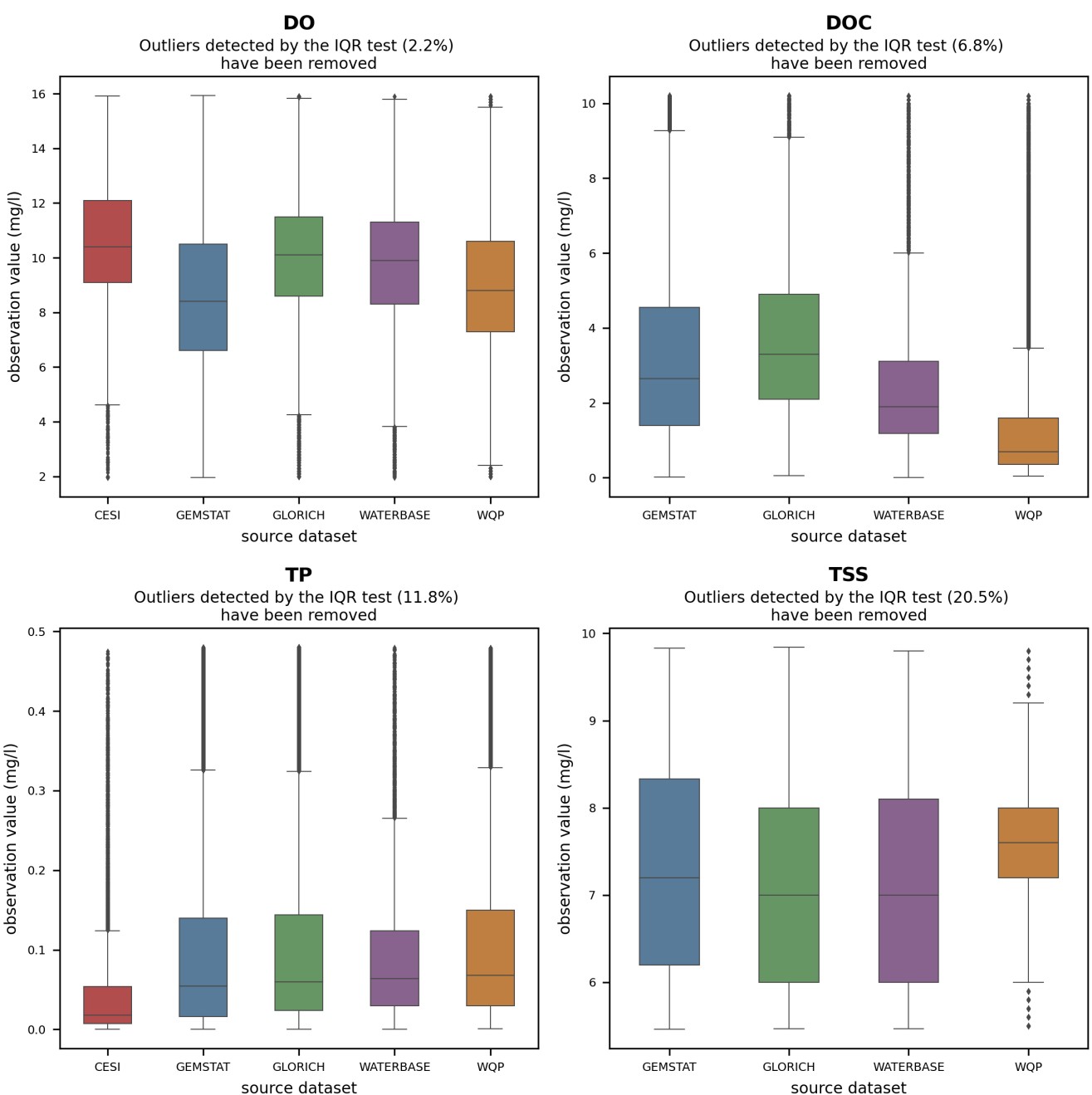

**Figure A1.** Box plot of observation values for dissolved oxygen (DO), dissolved organic carbon (DOC), total phosphorus (TP) and total suspended solids (TSS). Outliers determined by the IQR test (Table 7) are not shown on the plot.

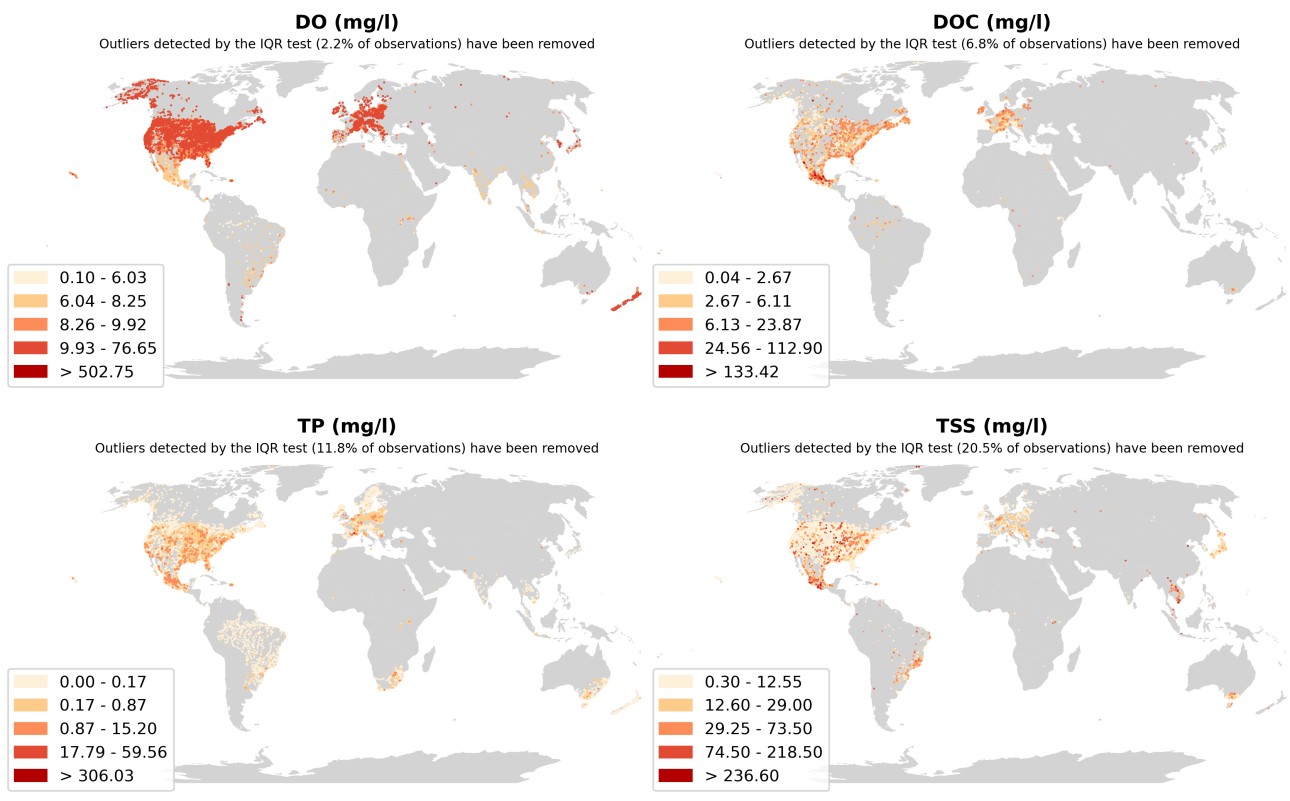

**Figure A2.** Spatial distribution of yearly median observation values for dissolved oxygen (DO), dissolved organic carbon (DOC), total phosphorus (TP) and total suspended solids (TSS). Outliers determined by the IQR test are not shown on the plot.

**Table A1.** Conversion procedures of source data units and chemical forms into their corresponding GRQA versions for all parameters.

| Parameter code | Source | Form | Source form | Unit | Source unit | Divisor | Multiplier | Conversion constant |
|---|---|---|---|---|---|---|---|---|
| TAN | CESI | N | NH3 | mg/l | MG/L | 17.031 | 14.007 | 0.822441 |
| NO3N | CESI | N | N | mg/l | MG/L | 1 | 1 | 1 |
| NO2N | CESI | N | N | mg/l | MG/L | 1 | 1 | 1 |
| TN | CESI | N | N | mg/l | MG/L | 1 | 1 | 1 |
| TDN | CESI | N | N | mg/l | MG/L | 1 | 1 | 1 |
| DO | CESI | O2 | O2 | mg/l | MG/L | 1 | 1 | 1 |
| pH | CESI | | | pH | PH UNITS | 1 | 1 | 1 |
| TP | CESI | P | P | mg/l | MG/L | 1 | 1 | 1 |
| TDP | CESI | P | P | mg/l | MG/L | 1 | 1 | 1 |
| TEMP | CESI | | | Deg C | DEG C | 1 | 1 | 1 |
| DC | GEMSTAT | C | C | mg/l | mg/l | 1 | 1 | 1 |
| DIC | GEMSTAT | C | C | mg/l | mg/l | 1 | 1 | 1 |
| DOC | GEMSTAT | C | C | mg/l | mg/l | 1 | 1 | 1 |
| POC | GEMSTAT | C | C | mg/l | $\mu$g/g | 1 | 1 | 1 |
| POC | GEMSTAT | C | C | mg/l | mg/l | 1 | 1 | 1 |
| TC | GEMSTAT | C | C | mg/l | mg/l | 1 | 1 | 1 |
| TIC | GEMSTAT | C | C | mg/l | mg/l | 1 | 1 | 1 |
| TOC | GEMSTAT | C | C | mg/l | mg/l | 1 | 1 | 1 |
| DKN | GEMSTAT | N | N | mg/l | mg/l | 1 | 1 | 1 |
| DON | GEMSTAT | N | N | mg/l | mg/l | 1 | 1 | 1 |
| NH4N | GEMSTAT | N | N | mg/l | mg/l | 1 | 1 | 1 |
| NH4N | GEMSTAT | N | NH4 | mg/l | mg/l NH4 | 18.039 | 14.007 | 0.776484 |
| NH4N | GEMSTAT | N | N | mg/l | $\mu$g/l | 1000 | 1 | 0.001 |
| NO2N | GEMSTAT | N | N | mg/l | mg/l | 1 | 1 | 1 |
| NO2N | GEMSTAT | N | NO2 | mg/l | mg/l NO2 | 46.005 | 14.007 | 0.304467 |
| NO2N | GEMSTAT | N | N | mg/l | $\mu$g/l | 1000 | 1 | 0.001 |
| NO3N | GEMSTAT | N | N | mg/l | mg/l | 1 | 1 | 1 |
| NO3N | GEMSTAT | N | NO3 | mg/l | mg/l NO3 | 62.004 | 14.007 | 0.225905 |
| NO3N | GEMSTAT | N | N | mg/l | $\mu$g/l | 1000 | 1 | 0.001 |
| PN | GEMSTAT | N | N | mg/l | mg/l | 1 | 1 | 1 |
| PON | GEMSTAT | N | N | mg/l | mg/l | 1 | 1 | 1 |
| PON | GEMSTAT | N | N | mg/l | $\mu$g/g | 1 | 1 | 1 |
| TDN | GEMSTAT | N | N | mg/l | mg/l | 1 | 1 | 1 |
| TKN | GEMSTAT | N | N | mg/l | mg/l | 1 | 1 | 1 |

| Parameter code | Source | Form | Source form | Unit | Source unit | Divisor | Multiplier | Conversion constant |
|---|---|---|---|---|---|---|---|---|
| TN | GEMSTAT | N | N | mg/l | mg/l | 1 | 1 | 1 |
| TON | GEMSTAT | N | N | mg/l | mg/l | 1 | 1 | 1 |
| DO | GEMSTAT | O2 | O2 | mg/l | mg/l | 1 | 1 | 1 |
| DOSAT | GEMSTAT | | | % | % | 1 | 1 | 1 |
| BOD | GEMSTAT | O2 | O2 | mg/l | mg/l | 1 | 1 | 1 |
| COD | GEMSTAT | O2 | O2 | mg/l | mg/l | 1 | 1 | 1 |
| pH | GEMSTAT | | | pH | — | 1 | 1 | 1 |
| DIP | GEMSTAT | P | P | mg/l | mg/l | 1 | 1 | 1 |
| TDP | GEMSTAT | P | P | mg/l | mg/l | 1 | 1 | 1 |
| TDP | GEMSTAT | P | P | mg/l | $\mu$g/l | 1000 | 1 | 0.001 |
| TIP | GEMSTAT | P | P | mg/l | mg/l | 1 | 1 | 1 |
| TP | GEMSTAT | P | P | mg/l | mg/l | 1 | 1 | 1 |
| TP | GEMSTAT | P | P | mg/l | $\mu$g/l | 1000 | 1 | 0.001 |
| TPP | GEMSTAT | P | P | mg/l | $\mu$g/g | 1 | 1 | 1 |
| TPP | GEMSTAT | P | P | mg/l | mg/l | 1 | 1 | 1 |
| TSS | GEMSTAT | | | mg/l | mg/l | 1 | 1 | 1 |
| TEMP | GEMSTAT | | | Deg C | °C | 1 | 1 | 1 |
| TEMP | GLORICH | | | Deg C | °C | 1 | 1 | 1 |
| pH | GLORICH | | | pH | | 1 | 1 | 1 |
| DO | GLORICH | O2 | O2 | mg/l | mg O2 L-1 | 1 | 1 | 1 |
| DOSAT | GLORICH | | | % | % | 1 | 1 | 1 |
| TSS | GLORICH | | | mg/l | mg L-1 | 1 | 1 | 1 |
| TC | GLORICH | C | C | mg/l | $\mu$mol L-1 | 1000 | 12.011 | 0.012011 |
| TIC | GLORICH | C | C | mg/l | $\mu$mol L-1 | 1000 | 12.011 | 0.012011 |
| DIC | GLORICH | C | C | mg/l | $\mu$mol L-1 | 1000 | 12.011 | 0.012011 |
| PIC | GLORICH | C | C | mg/l | $\mu$mol L-1 | 1000 | 12.011 | 0.012011 |
| TOC | GLORICH | C | C | mg/l | $\mu$mol L-1 | 1000 | 12.011 | 0.012011 |
| DOC | GLORICH | C | C | mg/l | $\mu$mol L-1 | 1000 | 12.011 | 0.012011 |
| POC | GLORICH | C | C | mg/l | $\mu$mol L-1 | 1000 | 12.011 | 0.012011 |
| TN | GLORICH | N | N | mg/l | $\mu$mol L-1 | 1000 | 14.007 | 0.014007 |
| TDN | GLORICH | N | N | mg/l | $\mu$mol L-1 | 1000 | 14.007 | 0.014007 |
| PN | GLORICH | N | N | mg/l | $\mu$mol L-1 | 1000 | 14.007 | 0.014007 |
| TIN | GLORICH | N | N | mg/l | $\mu$mol L-1 | 1000 | 14.007 | 0.014007 |
| DIN | GLORICH | N | N | mg/l | $\mu$mol L-1 | 1000 | 14.007 | 0.014007 |
| TON | GLORICH | N | N | mg/l | $\mu$mol L-1 | 1000 | 14.007 | 0.014007 |

| Parameter code | Source | Form | Source form | Unit | Source unit | Divisor | Multiplier | Conversion constant |
|---|---|---|---|---|---|---|---|---|
| DON | GLORICH | N | N | mg/l | $\mu$mol L-1 | 1000 | 14.007 | 0.014007 |
| PON | GLORICH | N | N | mg/l | $\mu$mol L-1 | 1000 | 14.007 | 0.014007 |
| TKN | GLORICH | N | N | mg/l | $\mu$mol L-1 | 1000 | 14.007 | 0.014007 |
| DKN | GLORICH | N | N | mg/l | $\mu$mol L-1 | 1000 | 14.007 | 0.014007 |
| NO3N | GLORICH | N | NO3 | mg/l | $\mu$mol L-1 | 1000 | 0.225905 | 0.000226 |
| NO2N | GLORICH | N | NO2 | mg/l | $\mu$mol L-1 | 1000 | 0.304467 | 0.000304 |
| NH4N | GLORICH | N | NH4 | mg/l | $\mu$mol L-1 | 1000 | 0.776484 | 0.000776 |
| TP | GLORICH | P | P | mg/l | $\mu$mol L-1 | 1000 | 30.973 | 0.030973 |
| TDP | GLORICH | P | P | mg/l | $\mu$mol L-1 | 1000 | 30.973 | 0.030973 |
| TPP | GLORICH | P | P | mg/l | $\mu$mol L-1 | 1000 | 30.973 | 0.030973 |
| TIP | GLORICH | P | P | mg/l | $\mu$mol L-1 | 1000 | 30.973 | 0.030973 |
| DIP | GLORICH | P | P | mg/l | $\mu$mol L-1 | 1000 | 30.973 | 0.030973 |
| NO3N | WATERBASE | N | NO3 | mg/l | mgNO3/L | 62.004 | 14.007 | 0.225905 |
| NO2N | WATERBASE | N | NO2 | mg/l | mgNO2/L | 46.005 | 14.007 | 0.304467 |
| NH4N | WATERBASE | N | NH4 | mg/l | mgNH4/L | 18.039 | 14.007 | 0.776484 |
| NH4N | WATERBASE | N | NH3 | mg/l | mgNH3/L | 17.031 | 14.007 | 0.822441 |
| NH3N | WATERBASE | N | NH3 | mg/l | mgNH3/L | 17.031 | 14.007 | 0.822441 |
| NH3N | WATERBASE | N | N | mg/l | ug/L | 1000 | 1 | 0.001 |
| TP | WATERBASE | P | P | mg/l | mgP/L | 1 | 1 | 1 |
| TSS | WATERBASE | | | mg/l | mg/L | 1 | 1 | 1 |
| TEMP | WATERBASE | | | Deg C | Cel | 1 | 1 | 1 |
| DOSAT | WATERBASE | | | % | % | 1 | 1 | 1 |
| DO | WATERBASE | O2 | O2 | mg/l | mg/L | 1 | 1 | 1 |
| DO | WATERBASE | O2 | O2 | mg/l | mgO2/L | 1 | 1 | 1 |
| BOD5 | WATERBASE | O2 | O2 | mg/l | mgO2/L | 1 | 1 | 1 |
| BOD7 | WATERBASE | O2 | O2 | mg/l | mgO2/L | 1 | 1 | 1 |
| CODCr | WATERBASE | O2 | O2 | mg/l | mgO2/L | 1 | 1 | 1 |
| CODMn | WATERBASE | O2 | O2 | mg/l | mgO2/L | 1 | 1 | 1 |
| DOC | WATERBASE | C | C | mg/l | mgC/L | 1 | 1 | 1 |
| DOC | WATERBASE | C | C | mg/l | mg/L | 1 | 1 | 1 |
| TOC | WATERBASE | C | C | mg/l | mgC/L | 1 | 1 | 1 |
| TOC | WATERBASE | C | C | mg/l | mg/L | 1 | 1 | 1 |
| pH | WATERBASE | | | pH | | 1 | 1 | 1 |
| TKN | WATERBASE | N | N | mg/l | mgN/L | 1 | 1 | 1 |
| TKN | WATERBASE | N | N | mg/l | mg/L | 1 | 1 | 1 |

| Parameter code | Source | Form | Source form | Unit | Source unit | Divisor | Multiplier | Conversion constant |
|---|---|---|---|---|---|---|---|---|
| TON | WATERBASE | N | N | mg/l | mgN/L | 1 | 1 | 1 |
| PON | WATERBASE | N | N | mg/l | mgN/L | 1 | 1 | 1 |
| TIN | WATERBASE | N | N | mg/l | mgN/L | 1 | 1 | 1 |
| TN | WATERBASE | N | N | mg/l | mgN/L | 1 | 1 | 1 |
| PC | WQP | C | C | mg/l | mg/l | 1 | 1 | 1 |
| DC | WQP | C | C | mg/l | mg/l | 1 | 1 | 1 |
| TC | WQP | C | C | mg/l | mg/l | 1 | 1 | 1 |
| DO | WQP | O2 | O2 | mg/l | mg/l | 1 | 1 | 1 |
| DOSAT | WQP | | | % | % saturatn | 1 | 1 | 1 |
| PIC | WQP | C | C | mg/l | mg/l | 1 | 1 | 1 |
| DIC | WQP | C | C | mg/l | mg/l | 1 | 1 | 1 |
| TIC | WQP | C | C | mg/l | mg/l | 1 | 1 | 1 |
| TAN | WQP | N | N | mg/l | mg/l as N | 1 | 1 | 1 |
| TAN | WQP | N | N | mg/l | mg/l as N | 1 | 1 | 1 |
| DIN | WQP | N | N | mg/l | mg/l as N | 1 | 1 | 1 |
| TIN | WQP | N | N | mg/l | mg/l as N | 1 | 1 | 1 |
| NO3N | WQP | N | N | mg/l | mg/l as N | 1 | 1 | 1 |
| NO3N | WQP | N | N | mg/l | mg/l as N | 1 | 1 | 1 |
| NO2N | WQP | N | N | mg/l | mg/l as N | 1 | 1 | 1 |
| NO2N | WQP | N | N | mg/l | mg/l as N | 1 | 1 | 1 |
| PON | WQP | N | N | mg/l | mg/l | 1 | 1 | 1 |
| DON | WQP | N | N | mg/l | mg/l | 1 | 1 | 1 |
| TON | WQP | N | N | mg/l | mg/l | 1 | 1 | 1 |
| POP | WQP | P | P | mg/l | mg/l as P | 1 | 1 | 1 |
| DOP | WQP | P | P | mg/l | mg/l as P | 1 | 1 | 1 |
| TOP | WQP | P | P | mg/l | mg/l as P | 1 | 1 | 1 |
| PN | WQP | N | N | mg/l | mg/l | 1 | 1 | 1 |
| TPP | WQP | P | P | mg/l | mg/l as P | 1 | 1 | 1 |
| TDP | WQP | P | P | mg/l | mg/l as P | 1 | 1 | 1 |
| TP | WQP | P | P | mg/l | mg/l as P | 1 | 1 | 1 |
| TP | WQP | P | P | mg/l | mg/l as P | 1 | 1 | 1 |
| TN | WQP | N | N | mg/l | mg/l | 1 | 1 | 1 |
| TDN | WQP | N | N | mg/l | mg/l | 1 | 1 | 1 |
| TN | WQP | N | N | mg/l | mg/l | 1 | 1 | 1 |
| POC | WQP | C | C | mg/l | mg/l | 1 | 1 | 1 |

| Parameter code | Source | Form | Source form | Unit | Source unit | Divisor | Multiplier | Conversion constant |
|---|---|---|---|---|---|---|---|---|
| DOC | WQP | C | C | mg/l | mg/l | 1 | 1 | 1 |
| TOC | WQP | C | C | mg/l | mg/l | 1 | 1 | 1 |
| BOD5 | WQP | O2 | O2 | mg/l | mg/l | 1 | 1 | 1 |
| BOD5 | WQP | O2 | O2 | mg/l | mg/l | 1 | 1 | 1 |
| pH | WQP | | | pH | std units | 1 | 1 | 1 |
| TSS | WQP | | | mg/l | mg/l | 1 | 1 | 1 |
| TEMP | WQP | | | Deg C | deg C | 1 | 1 | 1 |

*Author contributions.* Holger Virro conceived the manuscript, conducted the data processing and scripting. All authors contributed to the development of the workflow and writing the manuscript.

*Competing interests.* The authors declare no competing interests.

*Acknowledgements.* This research was funded by Mobilitas+ programme grant no. MOBERC34, Marie Skłodowska-Curie Actions individual fellowship under the Horizon 2020 Programme grant agreement number 795625, grant PRG352 from the Estonian Research Council,
NUTIKAS programme and the Dora Plus PhD student mobility scholarship number 36.9-6.1/1124 of the Archimedes foundation and European Regional Development Fund (EcolChange Centre of Excellence). Holger Virro is also thankful for technical support from the Yale Center for Research Computing support and the High Performance Computing Center of University of Tartu.

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
