# Peer review of "GRQA: Global River Water Quality Archive"

_Earth System Science Data, 2021_

## Author Comment (AC1)

Reply letter to the first review of the preprint submitted to Earth System Science Data (ESSD) entitled: "GRQA: Global River Water Quality Archive".

The authors' **answers** marked bold are given below the reviewer's comments. The updated version of the paper with changes highlighted in yellow can be found at the end of the document.

**General comment**

This manuscript present workflow and data for harmonizing and joining several major water quality data basis on international and national level to a consistent dataset. The authors carefully describe their workflow and present selected examples on the spatiotemporal coverage of their data sets and suggest further improvements. This is a good
example and a showcase what problems scientists have to face when joining datasets. The authors are very careful in making their workflow reproducible for scientists who want to work with the data, e.g. by flagging but not removing outliers. The resulting dataset is unique and very helpful for water quality researchers. I would consider the data as high quality with a few drawbacks mentioned below. While I very much appreciate the work the authors put into the data, I have some concerns regarding the manuscript itself. Overall I found the introduction rather weak. The authors build the justification for their work mainly on water quality modelling with a special focus on machine learning. For me this is to narrow. Any research rely on data, be it the mere statement of a global or regional statistics of a certain constituent, data-driven exploration of controlling factors behind observed patterns or statistical or even mechanistic models. I moreover miss a reference to the need for open data and the FAIR principles. Considering the workflow I greatly miss the handing of values below the limit of detection/ quantification. This is a major issue for all water quality studies and is left entirely open here. I hope the authors can make use of these general comments and the more detailed specific comments below.

**Answer:** Thank you for the useful comments! We addressed most of the issues under the specific comments (see below). We have now referred to the FAIR Data Principles when discussing the insights gained about the importance of metadata and usability of large-scale water quality data after compiling GRQA (line 67-69). We also processed the data again, so that observation values below and above detection limit were retained and flagged according to how they appeared in source data (see below). We reduced the focus on machine learning by broadening the value of water quality analysis in the introduction (line 16-22) and specifying that encouraging the uptake of open data policies by institutions collecting water quality data would still be the main way of improving the spatiotemporal coverage, rather than ML methods (line 375-376 & 381-383).

In addition, we included an extensive data catalog with maps showing the spatiotemporal coverage and graphs describing the distribution of all 42 parameters as supplementary material of the paper. The data catalog can be viewed at
https://drive.google.com/file/d/1UvI5tmeXff4Fwgard5xt0hl9SIh2rdra/view?usp=sharing.

**Specific comments**

**Abstract**
Line 1-4: I find the introduction rather weak. Some more specific words would be helpful to define the problem, to say what is already there and to what extent this manuscript is going beyond the status quo.

**Answer:** We added a small section to the beginning of the abstract addressing this (line 1-5).

Line 2: It is not immediately clear that "current" study is referring to this manuscript. Consider rewording.

**Answer:** Replaced that part of the sentence into the full name of the dataset (line 6).

**Introduction**

Line 13: I disagree here. Why is water quality modelling an integral part of ecosystem health monitoring? Why starting directly with modelling? Models need a conceptual basis that arises from data and interpretation of mechanisms behind the observations. I think there is great value in the data analysis itself before making the step to modelling.

L34ff: I see the point for putting emphasis on machine learning approaches. However, this seems to be the major justification for large-scale or large-sample datasets and the major outlet of these data. Here I disagree - ML is one possibility but surely not the only justification for the need for water quality data with a wide spatial and temporal coverage.

**Answer:** These are definitely valid remarks. Indeed, the focus might seem too skewed towards modeling and ML, rather than other methods of analysis. The initiative to focus on modeling came from the advent of large-scale streamflow modeling studies (e.g. Gudmundsson and Seneviratne, 2015). We saw water quality as complementary to streamflow models, as it would allow for large-scale nutrient or sediment runoff modeling. However, as we realized during working on this dataset, further improvements to the spatiotemporal coverage of global water quality data are still needed before modeling becomes more viable. We rephrased some of the sentences and added more wider context to the beginning of the introduction (line 16-22).

Line 21: What model inputs do you mean here? Please specify. Is it rather calibration data? Or input in terms of drivers (such as precipitation drives a rainfall-runoff model)? Or do you mean parameterization for processes such as a nutrient uptake?

**Answer:** The wording was indeed confusing here. We meant calibration and validation data in the form of water quality observations used when developing the model and verifying its performance. We have specified this now in text (line 30-31).

Line 28-33: Here you seem to jump to hydrological models (quantity, not quality) without further mentioning that. To what extent can this be transferred to water quality models?

**Answer:** Here, we referred to both quantity and quality in the sense that some issues regarding scaling the models are similar regardless of the phenomena that is being modeled, e.g. the amount of location-specific calibration and variables needed to upscale the model from the catchment level to a larger scale. We made the wording more clear (line 38-39).

Line 40: Do you really mean large-scale here? Or rather a wide spatial coverage?

**Answer:** Indeed, we meant large-scale in the sense of modeling on a continental or global level. As "large-scale" is commonly used in the context of spatial extent in hydrological modeling studies, we did not feel the need to specify here.

**Data**

Line 71f: What dataset "that had been previously collected from the other sources" are you referring to? Is that showing up later in the manuscript? Maybe introduce that in the preceding section?

**Answer:** We meant that out of the total selection of parameters included in GRQA only eight were available in CESI. The wording was poor on our part, so that has now been fixed (line 82-83).

Table 2: Would it make sense to state the date of download here? The databases are still actively fed with new data, right?

**Answer:** Good suggestion! We added the date of dataset retrieval to the caption of Table 1. The table has now been updated with additional information about site count range, mean time series length per site and mean number of observations per site. Other than GLORICH, which is a static dataset, they are indeed being updated with WQP collecting data on a daily basis and the others having irregular updates every once in a while. We also had date of retrieval in the references previously.

**Method**

I miss a description how the detection limit was handled. This is very crucial. Often numbers such as <0.01 mg/L are given. For some constituents this is rather the rule than the exception.

**Answer:** Thank you for pointing this out! Our original thought process was that keeping these values would result in having a lot of records with the same default value (e.g. 0.01) in the data, which would affect potential modeling efforts (e.g. overfitting to the lower value range). However, we agree that by keeping values flagged as below detection limit in source data can provide useful information in the case of certain parameters. We added these values to GRQA along with an additional column (*detection_limit_flag*) showing whether a particular value was below (<) or above (>) detection limit in the source data and mentioned this in line 183-184 and added this column description to Table 6.

Line 120: Are outliers just detected (flagged?) or also removed? Why are time series characteristics derived before removing duplicates? Are duplicate station or duplicate samples ment?

**Answer:** As noted in section 3.2, the outliers were indeed flagged and not removed. In paragraph 3.3 "duplicate observations" refers to samples rather than stations. Since we did not remove any of the potentially duplicate observations and left determining which of the detected matches are "true duplicates" that should be merged up to the end user, we also did not remove these values for statistical purposes.

Table 4: For me this table does not make sense. Why not stating all conversion information but only an example? The meaning of x1 to x2 should be mentioned in the header.

**Answer:** The table was intended to be just an example of the conversion of four different forms of nitrogen parameters, which were the ones most commonly in need of conversion in the source datasets. All other conversions made are given per dataset in the lookup table files (*\*_code_map.csv*) also used for parameter harmonization and mentioned in section 3.1 *Parameter harmonization*, so they can be found in the metadata package. We have added

the full list of unit conversion procedures in the appendix (Table A1) and referred to that in line 171.

Line 155: Surely this decision was made wisely. However, it is not clear for the reader why one km is the threshold here. Can you explain that? In a headwater catchment of 2 km2 size a shift of the sampling stations by 1 km would probably not treated as one joint size, right?

**Answer:** We looked for similar studies dealing with merging datasets and duplicated stations for determining the search radius. It appears that there is no consensus and the assessment of spatial proximity depends on the subjective threshold set by authors. For example, the GSIM streamflow dataset (Do et al., 2018) used a radius of 5 km for selecting potential duplicate gauging stations. We chose 1 km to avoid having too many false positives (e.g. in the case of small headwater catchments, as you mentioned) to evaluate in the second stage of deduplication (RMSE calculation). We have added an explanation for our radius distance in line 247-251.

Line 206: This is maybe not the best example/ justification. If an agricultural spill has a long-lasting effect on water quality, it would not be an outlier but create more than one elevated values. Consider adjusting the example here.

**Answer:** Thank you for the suggestion. We rephrased the sentence and added a better example (line 227-230).

Line 210ff: What about physical impossible values? Negative concentrations, water temperature above 100°C or concentrations above the solubility of a constituent? Would it make sense to check and remove those as (if they occur) they will influence the percentiles/ quartiles?

**Answer:** Please see next answer.

Figure 7: Here I realize the point of the physical possible range of parameters. Oxygen can, to the best of my knowledge not be much higher than 15 mg/L - sure there can be oversaturation. But I would claim that 502.75 mg/L is impossible. Would those value not hamper your outlier statistics?

**Answer:** Very good point!. There are certainly values that were flagged as outliers that are extremely high and "illogical" because of some data entry or equipment malfunction errors, rather than "natural" outliers (e.g. because of spill events), which also affect the statistical calculations (Fig. 7). Negative values were removed as reported in section 3.1 *Observation data filtering* (line 182). We did try to come up with ways to determine thresholds above which a value can be considered physically impossible and for parameters like temperature and pH something like you suggested could work. For chemical concentrations, however, perhaps a more sophisticated procedure based on freshwater quality guidelines (Enderlein et al., 1996) would be needed. We considered this to be out of the scope of our robust outlier detection procedure, but we have now added this as a suggestion when discussing the limitations of GRQA in section 5.1 *Skewness of observation values* (line 349-352). As we also discussed certain aspects of metadata quality in the available water quality datasets in our paper, the occurrence of these potentially faulty values can provide some context about the state of water quality data in general, if only to point out what could be avoided when collating such datasets in the future.

**Results**

Table 6: Attribute parameters such as "upstream basin area" and "Drainage region" have not been mentioned before. However, this is a very crucial information for researchers working with the data. Choosing sites according to their catchment sizes is common with the consequence that this information needs to be reliable. Was that just taken over from the original datasets?

**Answer:** We agree that the catchment size is important information. However, in this study we were only able to use the information that came from source data and was only extracted if available. We added a reference to this in line 201-202. We could not add the missing catchment sizes because that would have required global level catchment extraction which is already another potential study in itself.

Table 7: Number of digits is relevant here! Is that fixed for all constituents to the same value? If yes, is should be e.g., 0.010 for median phosphorous, not 0.01. This decision should be part of the text.

**Answer:** Thanks for noticing. The trailing zeros seemed to have been truncated when calculating the median values. It is now fixed (Table 7).

Line 267: Table reference is missing.

**Answer:** Thank you for pointing that out. We have now added the missing reference (line 293).

Line 291ff: Usually concentrations are considered to be lognormally distributed. A skewness does thus not purely result from outliers. You should mention that non-normal distribution rule here.

**Answer:** Good point, which we now have added to this section (line 319-320).

Figure 4 is less informative than figure 3 but basically shows the same. For me this is not necessary. Potentially both figures can be combined.

**Answer:** We moved the figure 4 to the appendix (Fig. A1). The figures were meant to be complementary with Fig. 3 showing the distribution of observation values in general as well as the proportion of observations per source dataset. On the other hand, Fig. 4 was meant to show the range of values within source datasets as there are differences between how the values are distributed depending on source.

Line 312: What is the question mark referring to?

**Answer:** Thank you for pointing that out. This was a broken reference that has now been fixed.

Line 330ff: This is a good idea to come up with suggestions for improving datasets. However, I strongly recommend to make this part of the problem definition in the introduction and the objectives of this study.

**Answer:** Thank you for the suggestion. We have addressed this in the introduction by listing this as one of the objectives in lines 67-69.

Line 352ff: The spatial lack of data is for me not directly connected to the adoption of machine learning. Yes, those techniques may can come up with an estimate. However,

data-driven models are only as good as their training data coverage. For a prediction under boundary conditions/ landscape properties that have not been sampled you will need a mechanistic approach. The whole discussion here is a bit misleading. I don't think the solution for spatial gaps is not gap filling but rather measurements or incentives for countries to share data.

**Answer:** We agree that increased measures for encouraging countries to publish their data in accordance with open data standards is still the best option when improving the spatial coverage of data. We changed this section, so that potential ML methods refer specifically to filling temporal gaps in time series, rather than spatial (line 381-383).

**References added to the paper:**

[revised manuscript text omitted]

---

## Author Comment (AC2)

Reply letter to the second review of the preprint submitted to Earth System Science Data (ESSD) entitled: "GRQA: Global River Water Quality Archive".

The authors' answers marked bold are given below the reviewer's comments. The updated version of the paper with changes highlighted in yellow can be found at the end of the document.

**General comments**

Virro et al. describe aggregating and harmonizing five national, continental and global datasets that can be used for global water quality models. Among the five selected datasets the GEMSTAT itself is a global database containing harmonized data from the contributing countries. The authors follow the ETL approach (Extract-Transform-Load) to bring the different data sets together. The methods are well described and data, metadata and scripts are available at the given websites. The authors suggest in their conclusions to transform the set of CSV files to a relational database in future what would further improve the usability of this dataset. I encourage the authors to do it. Also I like the idea to develop an online dashboard for GRQA. From a global modelers perspective the derived GRQA can be easily used. The authors selected 42 specific parameter relevant for modelling nutrients (water temperature, oxygen, nitrogen, phosphorous, carbon compounds …). Still, GRQA cannot solve the problem of data scarcity in Africa, Asia and South America and also the suggested machine learning cannot help here.
In general, the paper is well written and supports the publishing of GRQA. One issue for me is that it was not mentioned that WATERBASE is already integrated in GEMSTAT and how the authors dealt with this. There must be a lot of duplicate data from this fact. But the given procedures should have found all the doubled data.
I support the publication of this paper. Some minor issues are listed below.

**Answer:** Thank you for the supportive opinion! Indeed, the GRQA cannot solve the problem of data scarcity in Africa, Asia and South America but is only one step towards this. We have now specified that encouraging the uptake of open data policies by institutions collecting water quality data would still be the main way of improving the spatiotemporal coverage, rather than ML methods (line 375-376 & 381-383). This paper will provide a base database to complement, but also the guideline how new water quality datasets should be created so that they can be most efficiently used with modern data science methods and ML. See response regarding WATERBASE vs GEMSTAT below.

In addition, we included an extensive data catalog with maps showing the spatiotemporal coverage and graphs describing the distribution of all 42 parameters as supplementary material of the paper. The data catalog can be viewed at
https://drive.google.com/file/d/1Uvl5tmeXff4Fwgard5xt0hl9SIh2rdra/view?usp=sharing.

**Specific comments**

line 55: please list the parameter used here "most important water quality parameters" or refer to your table 7

**Answer:** We added a reference to Table 7 (line 65).

line 71: "… only eight parameter matched the data set" – you mean one of the 42 selected parameters? Pls clarify.

**Answer:** We meant that out of the total selection of parameters included in GRQA only eight were available in CESI. The wording was poor on our part, so that has now been fixed (line 82-83).

line 73 – 74: please give this numbers (how many parameter matched the set, site count range, mean time series length per site, average number of observations per site) for all data sets in a table and refer only to the parameters that were used

**Answer:** Thank you for the suggestion. We updated the table with this information (Table 1).

line 99: "WATERBASE has the shortest timeseries ... 2008 - 2018" - as far as I remember there are nutrient data available starting with 1992; please check again the disaggregated data; e.g. see the graphs given under this link
https://www.eea.europa.eu/data-and-maps/daviz/rivers-nutrient-trend-4#tabdashboard-01
The graph is based on data from WATERBASE.

**Answer:** The time period shown for WATERBASE in our paper refers to the time series length of the 15 parameters that were extracted and included in GRQA. For these parameters the earliest observations are from 2008.

line 125: "introduction: nutrients, carbon, sediments and oxygen"-Please refer to table 7 here.

**Answer:** Reference now added (line 141).

line 217: I understood that WATERBASE was included into GEMSTAT; please see https://www.waterandchange.org/en/european-water-quality-monitoring-data-in-gemstat-databaseundergoes-major-update/
Also here the time frame is given with 49 years for WATERBASE; please check and write a short paragraph how you dealt with this issue.

**Answer:** A valid remark. We became aware of the inclusion of WATERBASE into GEMSTAT some time after we had started working on compiling GRQA. However, only sites with more than three years of data were included in this update. As mean time series length per site was only 1.4 years in WATERBASE, a significant number of sites were left out, which is why we decided to include WATERBASE separately in GRQA. Although it is likely that there were many observations, which appeared both in GEMSTAT and WATERBASE, the duplicate detection procedure we implemented should have identified WATERBASE observations also appearing in GEMSTAT. We added an explanation about WATERBASE inclusion in line 112-117.

The time frame given for WATERBASE in our paper refers to the time series length of the 15 parameters that were extracted and included in GRQA, as mentioned in a previous comment.

Figure 2: Why is the percentage of observations (1898 - 1970) not given her? If no data exist before 1970 it should be mentioned here. The colors in the legend don't correspond to the colors in the plot. E.g., I can only guess what dark green is. Also I wonder what I can learn from this plot. Maybe the authors change this plot and show the time series available per continent instead. That is actually where I'm interested in as global modeler.

**Answer:** Only seven observations (0.0000169%) existed for DOC in the 1968–1970 period, so the number seemed to get lost due to rounding. We added this detail under the plot. We also changed the design of the plot to a more clear line plot. This figure was meant to show the temporal distribution of values per source, since depending on the dataset the distribution is different. For example, GEMSTAT has more observations from the more recent time period, while WQP contains quite a lot of data also from the earlier years, which can be important to know when studying long term trends.

Unfortunately, we were not able to produce the continental distribution plots due to that information not being available in the source data. This was specifically an issue with GEMSTAT and GLORICH, since in the case of CESI, WATERBASE and WQP the continent can be assumed to be the same for all sites. As not all observations had country of origin reported in GEMSTAT and GLORICH, they could not be linked to a specific continent and more complex spatial queries would have been needed to generate continental statistics. However, we added extensive data catalogue as an appendix with graphs and maps for temporal and spatial coverage of every variable. This should help potential users to get a better overview of the data before downloading it.

line 222-228: I wonder how can the same station be reported to different databases with different position information? In case it is the same station then both stations should contain the same time series? And how are you dealing with stations that are very dense (<1km) but not the same? For example at a tributary that is going parallel and there is one station in the tributary and one in the main stream before tributary is coming in. Did you find such cases and how did you deal with it?

**Answer:** These cases could be due to differences in rounding of coordinates or when a station's location has shifted over time, so that it appears in another dataset under a different ID and slightly different coordinates. Indeed, those station pairs should have the same observation values, so they would be detected by the RMSE calculation procedure and collected to the corresponding metadata file (*_dup_obs.csv*).

Regarding nearby stations, which are actually independent. For these stations, the second stage of duplication detection (RMSE calculation) should have identified them as separate time series, because only cases where RMSE=0 were flagged as potential duplicates. There were also stations that were closer than 1 km within the same source dataset, but we considered those stations to have been validated by the corresponding institutions (e.g. ICWRGC in the case of GEMSTAT). We also checked some of these nearby stations present within the same source data and based on the location and metadata (e.g. station name/description) it seemed like they had been placed there for a reason, e.g. to monitor water quality both upstream and downstream from a town or facility. Therefore, we decided to apply the duplication detection procedure only to stations from different source datasets.

line 296-297: It is not clear to me why the IQR test outliers are removed from the plot? For illustrative purpose – what does it mean? The plots would not look much different wouldn't they? So the outliers are not shown or are removed from the data? And actually for TSS the outliers can be really data e.g. before the crest of a flood or at the beginning of discharge reach a region specific threshold TSS can get extraordinary values.

**Answer:** As mentioned in section 3.2 *Outlier flagging*, no outliers were removed from the GRQA and a column indicating their status was added instead. However, including them in the plot would have seriously affected the visibility of the rest of the data points due to the

extreme skewness. The skewness of the data can be seen when examining the maximum values given in the summary statistics (*_processed_stats.csv*) found in the metadata, where in some cases the maximum value of a parameter exceeded that of the median by several orders of magnitude. It is likely that the most extreme outliers are caused by data entry errors (incorrect unit, e.g. reported in µg/l instead of mg/l etc) or equipment malfunction. Regarding TSS, it is true that in certain instances the concentration can be significantly greater than the median value, but similar to other parameters, there were some extremely high values, which would have affected the illustration. For example, in the case of TSS values in WQP (*WQP_processed_stats.csv*) the maximum (7700 mg/l) greatly exceeded the median (7.6 mg/l) as well.

Figure 4: e.g. TSS – 18.9% outliers removed from the plot – this 18.9% refers to all data? So it could be that 18% outliers are in GEMSTAT and the rest in the other datasets? Do I understand it right? please clarify in the figure text.

**Answer:** Yes, the percentage of outliers refers to the whole dataset. As the same percentage of outliers per parameter is given in Table 7, we did not feel the need to specify that again here and instead added a reference to Table 7 under this figure.

line 307 – 313: Please delete this paragraph. The focus of this paper is on merging data together rather than assessing the content. Therefore I would remove lines starting with "DOC concentrations are lower ..." and ending with "point sources". Even if Figure 7 is interesting from the content perspective I would remove it from this paper because it is not the main focus. In my opinion it is sufficient to mention that this kind of plots are available for all 42 parameters in the GRQA dataset.

**Answer:** We removed this part and moved the example plot to the appendix (Fig. A2).

line 320: What do the authors mean by "using the wrong form"? This is not clear to me. You should have checked the forms before using it in your codes and I'm sure you did. So how can it happen that a wrong form was used? Please explain.

**Answer:** Please see the next answer.

line 322: Why are units in GLORICH in µg/l a problem during conversion process? The authors transformed it to mg/l. I cannot see a problem here. Please clarify the discussion.

**Answer:** This was poor wording on our part. As mentioned in section 3 *Unit conversion*, not all parameters in source data had information about in which form (e.g. NO3 vs N) they were reported available, so assumptions had to be made about the chemical form when converting the units using the molecular mass. We did try to find additional information about these missing forms from proxy sources, e.g. in the case of GLORICH another dataset, where the author of GLORICH was involved with and that had metadata about the chemical forms available (Börker et al., 2020). These references have been included in the *form_ref* column in corresponding lookup tables (*_code_map.csv*) used during the harmonization stage. However, if the assumption made based on this limited ancillary information was incorrect then the conversion would have been affected as well. We have now added this explanation to the unit conversion description (line 160-162) and fixed the wording in line 340-341.

line 352: I doubt that ML methods alone can help to fill the gaps in Africa and Asia without support of any measurements in the regions. So, the argument given in line 359

contradicts the statement given in line 352. Maybe a combination of remote sensing techniques and water quality modelling (including ML) could help. Please revise.

**Answer:** We agree that increased measures for encouraging countries to publish their data in accordance with open data standards is still the best option when improving the spatial coverage of data. We changed this section, so that potential ML methods refer specifically to filling temporal gaps in time series, rather than spatial.

line 380: "The dataset is expected to have yearly updates after publishing" – are you sure that this is realistic? I would rather advise to remove this sentence. It is not necessary to state something like this in a paper, you can just do it if the capacity is available.

**Answer:** Valid remark. We removed this sentence as it is unknown if it will be viable.

line 383 – 384: I like very much the idea to put GRQA in a database and to make it accessible via an online dashboard.

**Answer:** Thank you for the comment. We will consider some options to improve the usability and have added some potential options to the conclusions (line 405-411).

**Technical corrections**
line 268: table reference is missing

**Answer:** Thank you for pointing that out. We have fixed it now (line 293).

line 269: given numbers of sites per parameter ("15 (POP) up to 90792 (pH)") don't correspondent with the numbers given in Table 7. Please clarify.

**Answer:** Thank you for pointing that out. These were likely placeholder values from some earlier version that are now fixed (line 294).

line 312: reference is missing

**Answer:** As we removed this section, this missing reference has also been removed.

**References added to the paper:**

[revised manuscript text omitted]